# Taf14 recognizes a common motif in transcriptional machineries and facilitates their clustering by phase separation

Guochao Chen[1,7], Duo Wang[1,2,7], Bin Wu[3,7], Fuxiang Yan[1,2], Hongjuan Xue[3], Quanmeng Wang[4], Shu Quan [5] & Yong Chen [1,6 ✉]

*Saccharomyces cerevisiae* TBP associated factor 14 (Taf14) is a well-studied transcriptional regulator that controls diverse physiological processes and that physically interacts with at least seven nuclear complexes in yeast. Despite multiple previous Taf14 structural studies, the nature of its disparate transcriptional regulatory functions remains opaque. Here, we demonstrate that the extra-terminal (ET) domain of Taf14 (Taf14$_{ET}$) recognizes a common motif in multiple transcriptional coactivator proteins from several nuclear complexes, including RSC, SWI/SNF, INO80, NuA3, TFIID, and TFIIF. Moreover, we show that such partner binding promotes liquid-liquid phase separation (LLPS) of Taf14$_{ET}$, in a mechanism common to YEATS-associated ET domains (e.g., AF9$_{ET}$) but not Bromo-associated ET domains from BET-family proteins. Thus, beyond identifying the molecular mechanism by which Taf14$_{ET}$ associates with many transcriptional regulators, our study suggests that Taf14 may function as a versatile nuclear hub that orchestrates transcriptional machineries to spatiotemporally regulate diverse cellular pathways.

[1] State Key Laboratory of Molecular Biology, National Center for Protein Science Shanghai, Shanghai Institute of Biochemistry and Cell Biology, Center for Excellence in Molecular Cell Science, Chinese Academy of Sciences, Shanghai 200031, China. [2] University of Chinese Academy of Sciences, Beijing 100049, China. [3] National Facility for Protein Science in Shanghai, Zhangjiang Lab, Shanghai Advanced Research Institute, Chinese Academy of Sciences, Shanghai 201210, China. [4] National Key Laboratory of Crop Genetic Improvement, Huazhong Agricultural University, Wuhan 430070, China. [5] State Key Laboratory of Bioreactor Engineering, East China University of Science and Technology, Shanghai 200237, China. [6] School of Life Science and Technology, ShanghaiTech University, Shanghai 201210, China. [7] These authors contributed equally: Guochao Chen, Duo Wang, Bin Wu. ✉email: yongchen@sibcb.ac.cn

*S*accharomyces cerevisiae Taf14, also referred to as Anc1, Taf30, Tfg3, and Swp29 in the literature, has essential roles in transcriptional regulation. Taf14 regulates diverse cellular processes, including cytoskeleton organization, heat response pathway, chromosome maintenance, cell cycle progression, DNA-damage response, and yeast metabolic cycle[1–5]. Consistent with its multifaceted functions, Taf14 physically associates with more than seven transcriptionally relevant complexes, including the chromatin remodeling complexes SWI/SNF, INO80, and RSC, the acetyltransferase complex NuA3, the general transcription factors TFIID and TFIIF, and the mediator complex[6–11]. How Taf14 associates with these complexes to coordinate their cellular functions in yeast remain elusive. It is difficult to dissect the specific contribution of Taf14 in each complex, as removal of Taf14 could affect functions of all Taf14-associated complexes. Thus, the detailed structural information of Taf14-mediated interactions is required to help us design the separation-of-function mutation to disrupt Taf14 association with one complex without affecting the others, thus precisely defining the role of Taf14 in a single complex.

Taf14 contains two separated domains, an N-terminal YEATS (Yaf9, ENL, AF9, Taf14, Sas5) domain and a C-terminal orphan domain. The YEATS domain is a recently identified histone reader domain that recognizes lysine acylation modification, including acetylation, crotonylation, and succinylation[3,12–14]. There are three YEATS domain proteins in *S. cerevisiae*, namely Taf14, Sas5, and Yaf9. Taf14 YEATS domain preferentially recognizes acetylated and crotonylated H3K9, and is essential for transcriptional regulation and DNA-damage response[3,12]. In *Homo sapiens*, YEATS-containing proteins include AF9, ENL, GAS41, and YEATS2. The YEATS domain of AF9 binds acetylated H3K9Ac and recruits H3K79 methyltransferase Dot1L to H3K9Ac-rich chromatins to modulate gene transcription[15]. The acetyllysine-binding of ENL YEATS domain initiates oncogenic transcriptional programs in acute myeloid leukemia[16]. GAS41 YEATS domain is vital for H2A.Z deposition and maintenance of ESC identity[17]. Collectively, these results highlight the essential functions of histone recognition by YEATS domains[14].

A previous bioinformatics study identified that the C-terminal orphan domain of Taf14 shared marginal sequence similarity to the ET domains from BET (Bromodomain and Extra-Terminal domain) family proteins[18]. BET-family proteins, including mammalian BRD2, BRD3, BRD4, and BRDT, contain the N-terminal tandem bromodomain and the C-terminal ET domain. The ET domain is also identified in many YEATS-containing proteins, including yeast proteins Taf14, Sas5, and human proteins AF9 and ENL[18]. Structure studies of ET domains from BRD3, BRD4, and AF9 have suggested that the ET domain is a general protein–protein interaction domain and recognizes a peptide substrate[19–21]. It is unclear whether Taf14 adopts a similar recognition mechanism to associate with different complexes. Interestingly, yeast growth assay showed that the Taf14 ET domain alone could completely rescue the growth defect in *taf14Δ* yeast strain, indicating a more critical role of the ET domain than the YEATS domain[2]. Thus, the ET domain of Taf14 may have additional functions other than solely mediating protein–protein interaction, which merits further investigation.

The RSC (Remodel the Structure of Chromatin) complex, the most abundant SWI/SNF-family chromatin remodeling complex in yeast, has critical roles in transcriptional regulation, DNA replication, and DNA repair[9,22]. Sth1 is the catalytic subunit of the RSC complex and directly interacts with most subunits in the complex[23]. Taf14 is an accessory subunit of RSC and is tethered to RSC through the interaction with the C-terminal region of Sth1 (residues 1183–1359)[2,24]. However, the exact role of Taf14 in the RSC complex remains to be determined.

In the present study, we seek to determine how Taf14 is associated with different transcriptional machineries and the roles of Taf14 in these complexes. As Sth1 is one of the best-characterized Taf14-binding partners, we use the Taf14–Sth1 complex as an example. We conduct biochemical binding assays followed by nuclear magnetic resonance (NMR) analyses and mutagenesis studies, which reveal the critical residues mediating the interaction between Taf14's ET domain (Taf14$_{ET}$) and Sth1's ET-binding motif (Sth1$_{EBM}$). Subsequent genetics analyses implicate the involvement of Taf14–Sth1 interaction in multiple cellular functions, including heat resistance and carbohydrate metabolism. Further biochemical and NMR studies of previously reported Taf14-binding partners lead to the identification of a common binding motif among these Taf14-binding proteins. Moreover, fluorescence imaging analyses reveal that Taf14$_{ET}$ can undergo liquid–liquid phase separation (LLPS) in vitro and partner binding strongly promotes LLPS of Taf14$_{ET}$. Thus, we have identified the specific region of Taf14 which can account for its previously disparate transcriptional regulation activities, and have uncovered a physiochemical mechanism that helps explain how this protein can function as a concentration-aggregating local hub of transcriptional activation.

## Results

**Taf14$_{ET}$ binds a short fragment of Sth1**. Previous studies have suggested that the Taf14 binds Sth1$_{1183–1359}$[2,24] (Fig. 1a). We first confirmed that Sth1$_{1183–1359}$ indeed directly interacted with GST-Taf14$_{174–244}$ (Taf14$_{ET}$) through GST pull-down assay (Fig. 1b). Then we generated a series of Sth1 constructs in the residues 1183–1359 range to assess their interactions with GST-Taf14$_{ET}$. We found that the bromodomain of Sth1 (Sth1$_{1248–1359}$) did not interact with Taf14$_{ET}$, while the Sth1 fragments comprising residues 1183–1240 and 1199–1225 were able to associate with Taf14$_{ET}$ (Fig. 1b). The interaction between Taf14$_{ET}$ and Sth1$_{1183–1240}$ was further confirmed by isothermal titration calorimetry (ITC) (Fig. 1c). Taf14$_{ET}$ binds Sth1$_{1183–1240}$ tightly, with a dissociation constant ($K_D$) around 55 nM. In addition, the hydrophobic contacts might have an essential role in mediating the Taf14$_{ET}$-Sth1$_{1183–1240}$ complex formation, because increasing the salt concentrations only slightly decreased the observed binding affinity and did not affect enthalpy change ($\Delta H$) (Fig. 1c). Hereafter, we abbreviate Sth1$_{1183–1240}$ as Sth1$_{EBM}$ (ET-binding motif).

We then used NMR spectroscopy to probe possible conformational changes of Taf14$_{ET}$ upon Sth1$_{EBM}$ binding by collecting the $^{15}$N-$^1$H heteronuclear single quantum coherence (HSQC) spectra. The HSQC spectra of $^{15}$N-Taf14$_{ET}$ revealed a concentration-dependent aggregation property of apo Taf14$_{ET}$ (Fig. 1d). The peaks of 0.05 mM Taf14$_{ET}$ were well-dispersed with some overlapping signals. When the Taf14$_{ET}$ concentration was increased above 0.2 mM, the spectra featured a cluster of intense and broad peaks in the center of the spectrum (Fig. 1d). In contrast, the HSQC spectra of the complex containing both $^{15}$N-labeled Taf14$_{ET}$ and unlabeled Sth1$_{EBM}$ exhibited good chemical shift dispersion and equal intensity with the expected number of amide NH peaks even at 0.5 mM protein concentration, implying a stable folded structure (Fig. 1e). We also used circular dichroism (CD) to characterize the structures of Taf14$_{ET}$, Sth1$_{EBM}$ and the Taf14$_{ET}$–Sth1$_{EBM}$ complex. Sth1$_{EBM}$ was in an almost entirely random-coil conformation, while both Taf14$_{ET}$ alone and in complex with Sth1$_{EBM}$ peptide showed similar spectra features representing a mixed α–β structure (Fig. 1f). These results indicate that Sth1$_{EBM}$-binding stabilizes Taf14$_{ET}$ conformation in solution.

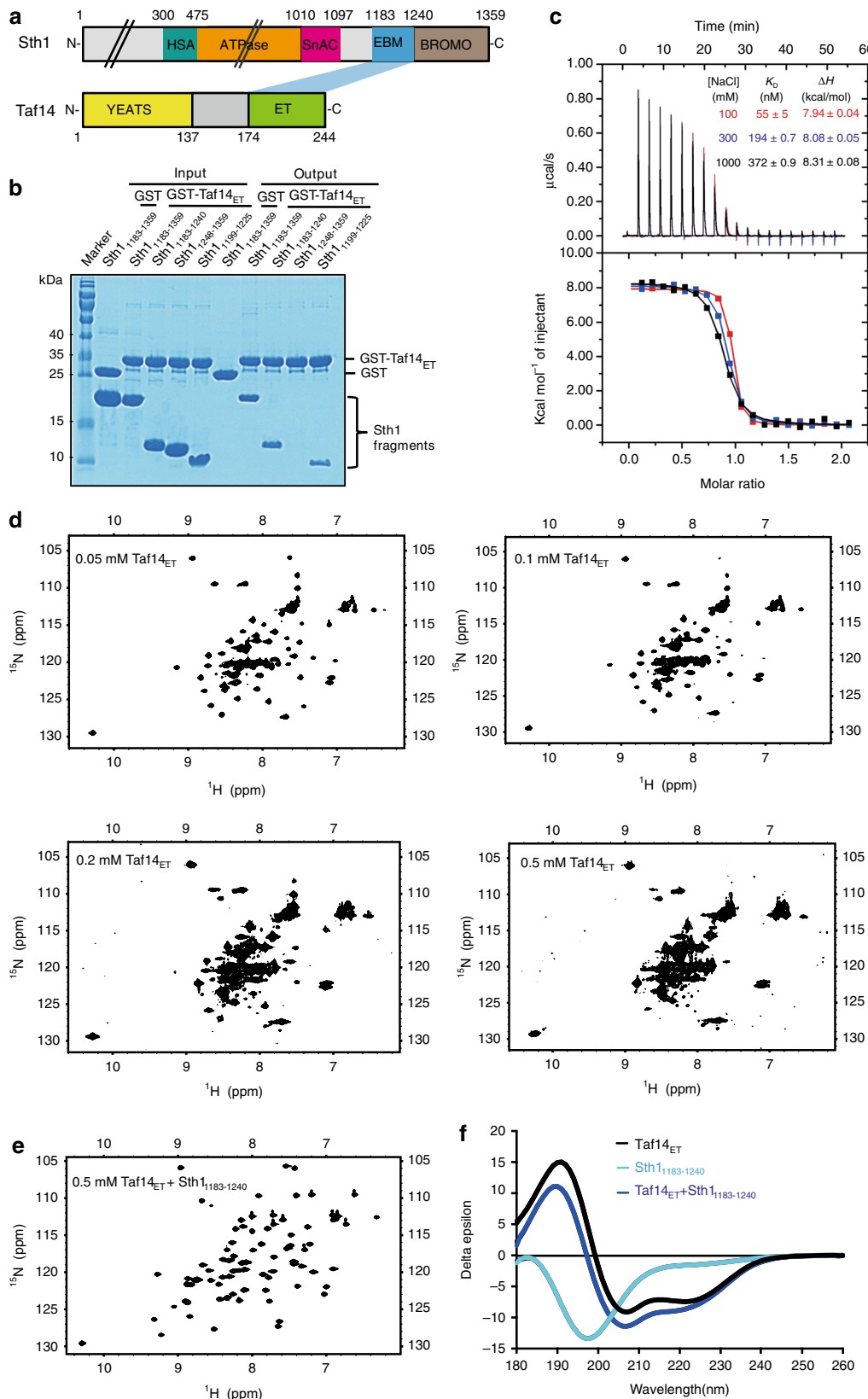

**Solution structure of Taf14$_{ET}$–Sth1$_{EBMC}$ reveals a compact fold**. To investigate the molecular mechanism through which Taf14$_{ET}$ recognizes Sth1$_{EBM}$, we determined the solution structure of the Taf14$_{ET}$–Sth1$_{EBM}$ complex using NMR (Table 1). We observed a plethora of conformational ensembles of Sth1$_{EBM}$,

suggesting that most regions of Sth1$_{EBM}$ are highly dynamic (Supplementary Fig. 1a, b). Only a 9-residue core fragment (Sth1$_{1203–1211}$; hereinafter referred to as Sth1$_{EBMC}$, Extra-domain Binding Motif Core) adopts an ordered structure, indicating that this motif directly interacts with Taf14$_{ET}$ (Supplementary Fig. 1a).

**Fig. 1 Taf14$_{ET}$ binds a short motif of Sth1. a** Domain organization of the ScSth1 and ScTaf14 proteins. HSA Helicase-SANT-Association domain, ATPase ATPase domain, SnAc Snf2 ATP coupling domain, EBM ET-Binding Motif, Bromo Bromodomain, YEATS Yaf9-ENL-AF9-Taf14-Sas5 shared domain, ET Extra-Terminal domain. **b** GST pull-down assays showed the interaction between GST-Taf14$_{174-244}$ and four different Sth1 fragments. GST-Taf14$_{174-244}$ could pull-down Sth1$_{1183-1359}$, Sth1$_{1183-1240}$, and Sth1$_{1199-1225}$, but not Sth1$_{1248-1359}$. **c** The interaction between Taf14$_{ET}$ (Taf14$_{174-244}$) and Sth1$_{EBM}$ (Sth1$_{1183-1240}$) was not affected by altering solution ionic strength, as shown by ITC measurements at three different salt concentrations. The upper panel shows the heat change upon titration; the lower panel shows the binding isotherm profile fit based on a "one binding site" model. The dissociation constant ($K_D$), enthalpy change ($\Delta H$), and their fitting errors were shown. **d** [$^{15}$N-$^1$H] HSQC of Taf14$_{174-244}$ at different concentrations indicated the concentration-dependent aggregation in solution. 0.05 mM, 512 scans; 0.1 mM, 128 scans; 0.2 mM, 128 scans; 0.5 mM, 32 scans. **e** [$^{15}$N-$^1$H] HSQC of Taf14$_{174-244}$ at 0.5 mM concentration in the presence of 0.75 mM unlabeled Sth1$_{1183-1240}$ indicated a well-folded structure. **f** Circular dichroism spectra of Taf14$_{174-244}$ (0.1 mg/mL), Sth1$_{1183-1240}$ (0.1 mg/mL), Taf14$_{174-244}$-Sth1$_{1183-1240}$ (0.15 mg/mL) in 2 mM HEPES, pH 7.0, 100 mM NaF buffer.

**Table 1 NMR and refinement statistics for the Taf14$_{174-244}$-Sth1$_{1183-1240}$ structure.**

|  | Taf14$_{174-244}$-Sth1$_{1183-1240}$ |
|---|---|
| **NMR distance and dihedral constraints** | |
| Distance constraints | |
| Total NOE | 3126 |
| Intra-residue | 643 |
| Inter-residue | |
| Sequential (\|i – j\| = 1) | 714 |
| Medium-range (\|i – j\| < 4) | 854 |
| Long-range (\|i – j\| > 5) | 915 |
| Hydrogen bonds | 70 |
| Total dihedral angle restraints | 142 |
| ϕ | 71 |
| ψ | 71 |
| **Structure statistics** | |
| Violations (mean and s.d.) | |
| Distance constraints (Å) | 0.015 ± 0.0013 |
| Dihedral angle constraints (°) | 0.24 ± 0.088 |
| Max. dihedral angle violation (°) | 3.80 |
| Max. distance constraint violation (Å) | 0.39 |
| Deviations from idealized geometry | |
| Bond lengths (Å) | 0.0146 ± 0.0003 |
| Bond angles (°) | 0.95 ± 0.017 |
| Average pairwise r.m.s.d.$^a$ (Å) | |
| Heavy | 0.60 ± 0.13 |
| Backbone | 0.30 ± 0.12 |
| Ramachandran plot statistics (%) | |
| Residues in most favored regions | 85.3 |
| Residues in additional allowed regions | 14.7 |
| Residues in generously allowed regions | 0.0 |
| Residues in disallowed regions | 0.0 |

$^a$r.m.s.d., root mean square deviation. Pairwise r.m.s.d was calculated among 20 refined structures for residues Taf14 (177–243) and Sth1 (1203–1211).

The Taf14$_{ET}$–Sth1$_{EBMC}$ complex forms a compact fold with three α-helices and three β-strands (Fig. 2a). The three α-helices (α1, α2, and α3) form a helical bundle, which is covered by the β-sheet composed of the two strands from Taf14$_{ET}$ (βA and βB) and one β-strand from Sth1$_{EBMC}$ (β1). This Sth1$_{EBMC}$ β1 strand is embraced by βB, α1, and α2 of Taf14$_{ET}$, forming an integral hydrophobic core in the complex (Fig. 2a).

The formation of the binary complex involves extensive interactions, with 782 Å$^2$ of burial surface area (Fig. 2b). The driving forces for the binding of Sth1$_{EBMC}$ to Taf14$_{ET}$ are hydrophobic contacts, which is consistent with the observation that the high affinity between Taf14$_{ET}$ and Sth1$_{1183-1240}$ can still be maintained with a 1 M salt buffer concentration (Fig. 1c). Four hydrophobic residues of Sth1$_{EBM}$ (L1204, V1206, I1208, and L1210) form extensive contacts with a panel of non-polar residues

from α1 and α2 of Taf14$_{ET}$ (Fig. 2c), thereby stabilizing the tertiary structure of the complex. Electrostatic interactions further stabilize the Sth1$_{EBMC}$–Taf14$_{ET}$ interaction. The side chains of Sth1$^{K1207}$ and Sth1$^{K1209}$ are oriented towards a negatively charged pocket of Taf14 comprising E216, E217, and E219, along with Sth1$^{K1203}$ close to Taf14$^{E223}$ (Fig. 2b). Thus, both hydrophobic contacts and electrostatic interactions ensure the specific recognition of Sth1$_{EBMC}$ by Taf14$_{ET}$.

**Structure-based mutations reduce the Taf14–Sth1 interaction.** To validate our structural model of the Taf14$_{ET}$–Sth1$_{EBM}$ complex, we examined whether mutations at the predicted interface could weaken the interactions between Taf14$_{ET}$ and Sth1$_{EBM}$. ITC analyses showed that alanine substitution of interfacial hydrophobic residues (L1204, V1206, I1208, and L1210) of Sth1$_{EBM}$ severely hindered interactions with Taf14$_{ET}$ ($K_D$ increased ~20−270 fold); the Sth1$^{I1208A}$ mutation caused the most pronounced disruption of the complex interface (Fig. 2d; Supplementary Fig. 2a). Simultaneous mutation of all four sites, L1204A/V1206A/I1208A/L1210A (Sth1$^{M4}$), completely abolished the interaction between Taf14$_{ET}$ and Sth1$_{EBM}$. Further ITC analyses showed that charge-flipping mutations of two positive-charge residues (K1207E and K1209E) also hindered Taf14$_{ET}$–Sth1$_{EBM}$ binding ($K_D$ increased 154- and 79-fold, respectively) (Fig. 2d; Supplementary Fig. 2b). Moreover, mutation of Taf14$_{ET}$ residues also weakened Sth1-binding. Arginine substitution of the non-polar F220 and I222 residues of Taf14$_{ET}$ substantially reduced the binding affinity with Sth1$_{EBM}$ without the disruption of the structural integrity of Taf14$_{ET}$ (Fig. 2d; Supplementary Fig. 2c, d). The combination of these two mutations completely abolished the Taf14$_{ET}$–Sth1$_{EBM}$ complex formation (Fig. 2d; Supplementary Fig. 2c).

Extending beyond two specific domains, we next explored whether the hydrophobic interfaces identified in the NMR structure of Taf14$_{ET}$–Sth1$_{EBM}$ complex also promoted the binding of the full-length Taf14 and Sth1 proteins. We generated yeast strains with a TAP-tagged Sth1 (Sth1$^{WT}$ or Sth1$^{M4}$) and an HA-tagged Taf14 (Supplementary Table 1). Co-immunoprecipitation (CoIP) experiments revealed that the quadruple mutant Sth1$^{M4}$-TAP severely reduced binding with Taf14-HA (Fig. 2e; Supplementary Fig. 2e). Silver staining of RSC complexes purified from yeast cells expressing the Sth1$^{WT}$ or Sth1$^{M4}$ proteins revealed identical banding patterns (Supplementary Fig. 2f), suggesting that the Sth1$^{M4}$ mutation specifically weakens the Sth1–Taf14 interaction rather than affecting the overall integrity of the RSC complex by additionally disrupting other Sth1-mediated interactions. Thus, our mutagenesis analyses support our conclusions from the structure model and demonstrate how hydrophobic and electrostatic residues mediate the interaction between Taf14$_{ET}$ and Sth1$_{EBM}$.

**The Taf14–Sth1 interaction regulates a plethora of pathways.** To reveal the functions of the Taf14–Sth1 interaction, we examined the phenotypes of yeast strains disrupted for the

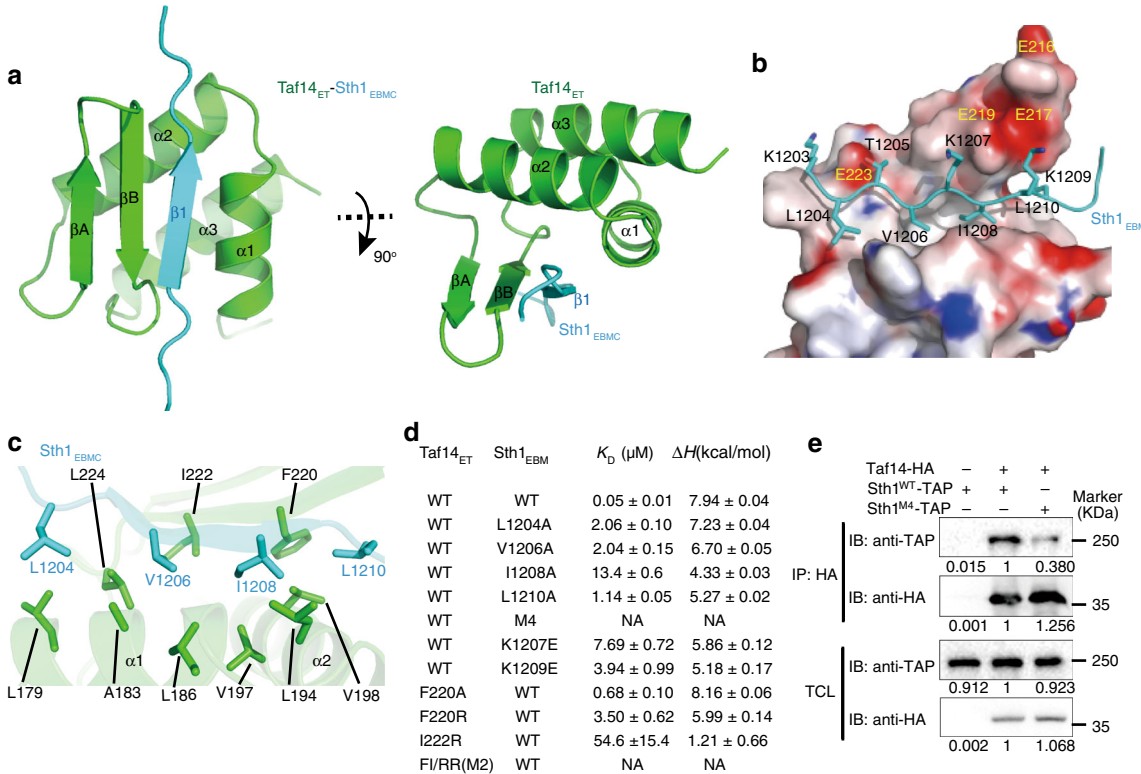

**Fig. 2 Structure of the Taf14$_{ET}$–Sth1$_{EBM}$ complex determined by NMR. a** Two orthogonal views of the Taf14$_{ET}$–Sth1$_{EBM}$ complex. Taf14$_{ET}$ is colored in green and Sth1$_{EBM}$ is colored in cyan. **b** The interface between Taf14$_{ET}$ and Sth1$_{EBM}$. Sth1$_{ET}$ is shown as a surface model colored according to its electrostatic potential (positive potential: blue; negative potential, red). The residues through which Sth1$_{EBM}$ interacts with the Taf14$_{ET}$ are presented as ball-and-stick models. The four acidic residues of Taf14$_{ET}$ are labeled. **c** Details of hydrophobic interactions. The critical residues are presented as ball-and-stick models. **d** ITC results showing that the targeted mutations decreased the affinity of the interaction between Taf14$_{ET}$ and Sth1$_{EBM}$. M4 stands for the Sth1$^{L1204A/V1206A/I1208A/L1210A}$ mutation. **e** Co-immunoprecipitation using an HA antibody showed that Sth1$^{M4}$ mutation weakened the interaction with Taf14. Taf14 was tagged with HA; Sth1 was tagged with TAP. TCL total cell lysate, IP immune-precipitation, IB immune-blot. Source data are provided as a Source Data file.

Taf14–Sth1 interaction. The *taf14Δ* and *taf14^M* (*taf14^M* is *taf14 F220R/I222R*) cells showed significant growth defects at various temperatures, different NaCl concentrations, and under all of the DNA-damage stress conditions tested (Supplementary Fig. 3a–c). These growth defects are not unexpected, as Taf14 is known to interact with at least seven different nuclear protein complexes[24]. Assuming that the *taf14Δ* and *taf14^M* cells are deficient for all Taf14-mediated interactions, we focused our attention on the impacts of the *sth1^M4* mutation, which specifically disrupts the Taf14–Sth1 interaction and allows us to precisely dissect the roles of Taf14 in the RSC complex.

Since *sth1* is an essential gene and *sth1Δ* is lethal for yeast, we transformed the diploid strain BY4743 (one allele of *sth1* deleted) with pRS313 plasmids carrying either *STH1* or *sth1^M4* alleles. The transformants were sporulated and subjected to tetrad dissection to isolate spore clones carrying *STH1* or *sth1^M4* plasmids for viability. The *sth1^M4* strains displayed strong heat-sensitivity and incapability for growth at both 34 and 37 °C, and showed negligible growth defects at 30 °C and below (Fig. 3a). Growth curves in liquid culture confirmed the observed phenotypes on plates (Supplementary Fig. 3d). The *sth1^M4* strains also failed to grow in the presence of high salt concentration (1 and 1.5 M NaCl), but grew normally at 0.2 and 0.5 M NaCl condition (Fig. 3b). These phenotypes were in sharp contrast to that of *taf14Δ* cells, which exhibited strong growth defects at all tested temperatures and NaCl concentrations (Supplementary Fig. 3a, b). Thus, the Taf14–Sth1 interaction is particularly impactful for the heat resistance and high osmotic stress responses of yeast cells. We also

found that the *sth1^M4* mutation made the strain slightly sensitive to hydroxyurea (HU) and phleomycin (Phle); no sensitivity to methyl methanesulfonate (MMS) was observed (Fig. 3c), which also stood out in comparison with the *taf14Δ* mutant showing strong sensitivity to all tested DNA-damage agents (Supplementary Fig. 3c). These results imply that Taf14–Sth1 interaction mildly affects specific DNA-repair processes.

To profile the impact of disrupting the Taf14–Sth1 interaction on the transcriptome, we performed RNA-seq of the strains expressing *STH1* and *sth1^M4*. The *sth1^M4* mutation affected gene expression globally: there were 180 significantly differentially upregulated genes and 202 downregulated genes (fold change > 2) (Fig. 3d). In addition, qPCR confirmed that *sth1^M4* increased or decreased the expression of the representative genes identified in the RNA-seq analysis, while the expression of a control gene (*Mgm1*) remained unchanged (Fig. 3e; Supplementary Fig. 3e). Some of the downregulated genes had functional annotations associated with heat-resistance responses (Fig. 3f), which may help explain our observation that the strain expressing the *sth1^M4* protein was sensitive to high but not low-temperature growth conditions (Fig. 3a). The gene ontology analysis also indicated that the downregulated genes were enriched for genes with annotations related to carbohydrate metabolism (Fig. 3f), raising the possibility that the Taf14–Sth1 interaction may regulate energy metabolism in yeast cells. This speculation was supported by our observation that the *sth1^M4* but not the wild-type strain showed growth defects when cultured using galactose or raffinose as the sole carbon source (Fig. 3g).

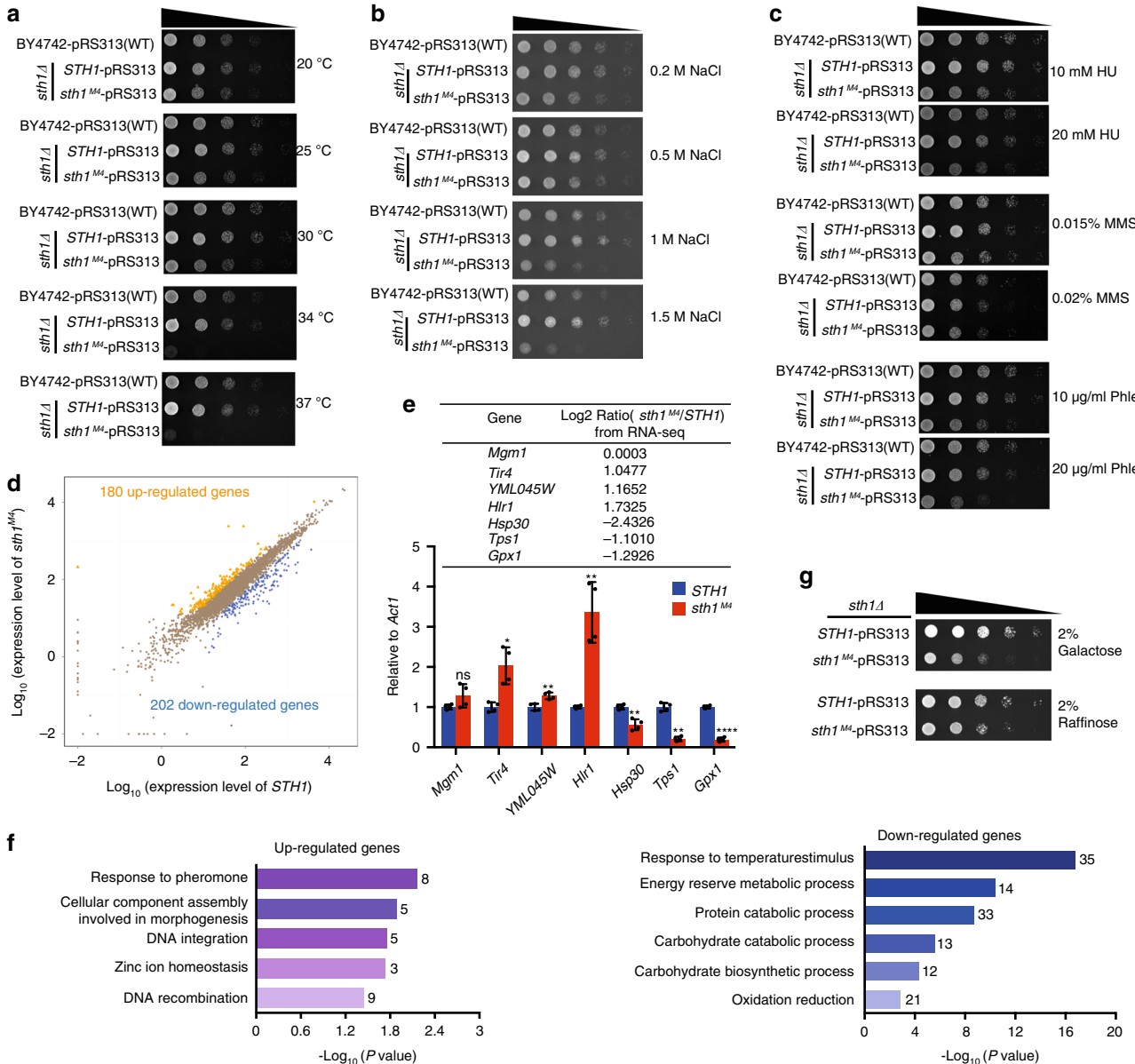

**Fig. 3 Genetic analyses of the Taf14–Sth1 interaction. a** Spotting assays with *sth1Δ* strains transformed with plasmids expressing full-length wild-type Sth1 or M4 mutant (L1204A/V1206A/I1208A/L1210A) at different temperatures. **b** Spotting assays with *sth1Δ* strains transformed with plasmids expressing full-length wild-type Sth1 or M4 mutant at different NaCl concentrations at 30 °C. **c** Spotting assays with *sth1Δ* strains transformed with plasmids expressing full-length wild-type or M4 mutant *STH1* on plates containing hydroxyurea (HU), methyl methanesulfonate (MMS), and phleomycin (Phle) at two different concentrations at 30 °C. **d** Scatter plot of log₁₀ normalized gene counts in the *STH1* and *sth1^M4* strains. 108 genes were significantly (>2-fold change) upregulated (orange) and 202 genes were downregulated (blue) in the *sth1^M4* strain. **e** The qPCR validation of the RNA-seq results. *Tir4*, *YML045W*, and *Hlr1* were upregulated, while *Hsp30*, *Tps1*, and *Gpx1* were downregulated in *sth1^M4* strains. The *Mgm1* transcription level is not regulated by Sth1. The change levels from RNA-seq were shown. Error bars represent standard deviations of four replicates. The relative expression level of wild-type *STH1* strain genes was set to 1. * for *P* < 0.05; ** for *P* < 0.01; *** for *P* < 0.001; **** for *P* < 0.0001; two-tailed Student's *t*-test. Data are presented as mean ± SD, *n* = 4. Source data are provided as a Source Data file. **f** Gene ontology analysis of the upregulated and downregulated differentially expressed genes. **g** Spotting assays of *sth1Δ* strains transformed with plasmids expressing full-length *STH1* or *sth1^M4* on plates with galactose or raffinose as the sole carbon source at 30 °C.

**Taf14_ET recognizes a conserved motif in diverse binding proteins**. After we dissected the structural basis of the Taf14 association with the RSC complex, we wondered how Taf14 integrates with other complexes. Pioneering studies have suggested that Taf14 interacts with the following regions: residues 311–550 of the Ino80 subunit of the INO80 complex, residues 96–360 of the Sas3 subunit of the NuA3 complex, residues 1381–1407 of the Taf2 subunit of the TFIID complex, and the Snf5 subunit of the SWI/SNF complex[8,24,25]. Building from these

reports, we used GST pull-down assays to confirm and further narrow down the regions through which Taf14_ET interacts with Ino80, Sas3, and Snf5 (Supplementary Fig. 4a–c). The resulting minimum fragments (Ino80_{368–390}, Snf5_{771–800}, Sas3_{111–129}, and Taf2_{1393–1406}) were confirmed by ITC, showing Taf14_ET binding with various affinities (Fig. 4a, d; Supplementary Fig. 4d, e).

HSQC spectra further showed that titration of these minimum peptides with ¹⁵N-labeled Taf14_ET induced similar chemical shift patterns as we had observed with the Taf14_ET–Sth1_EBM complex

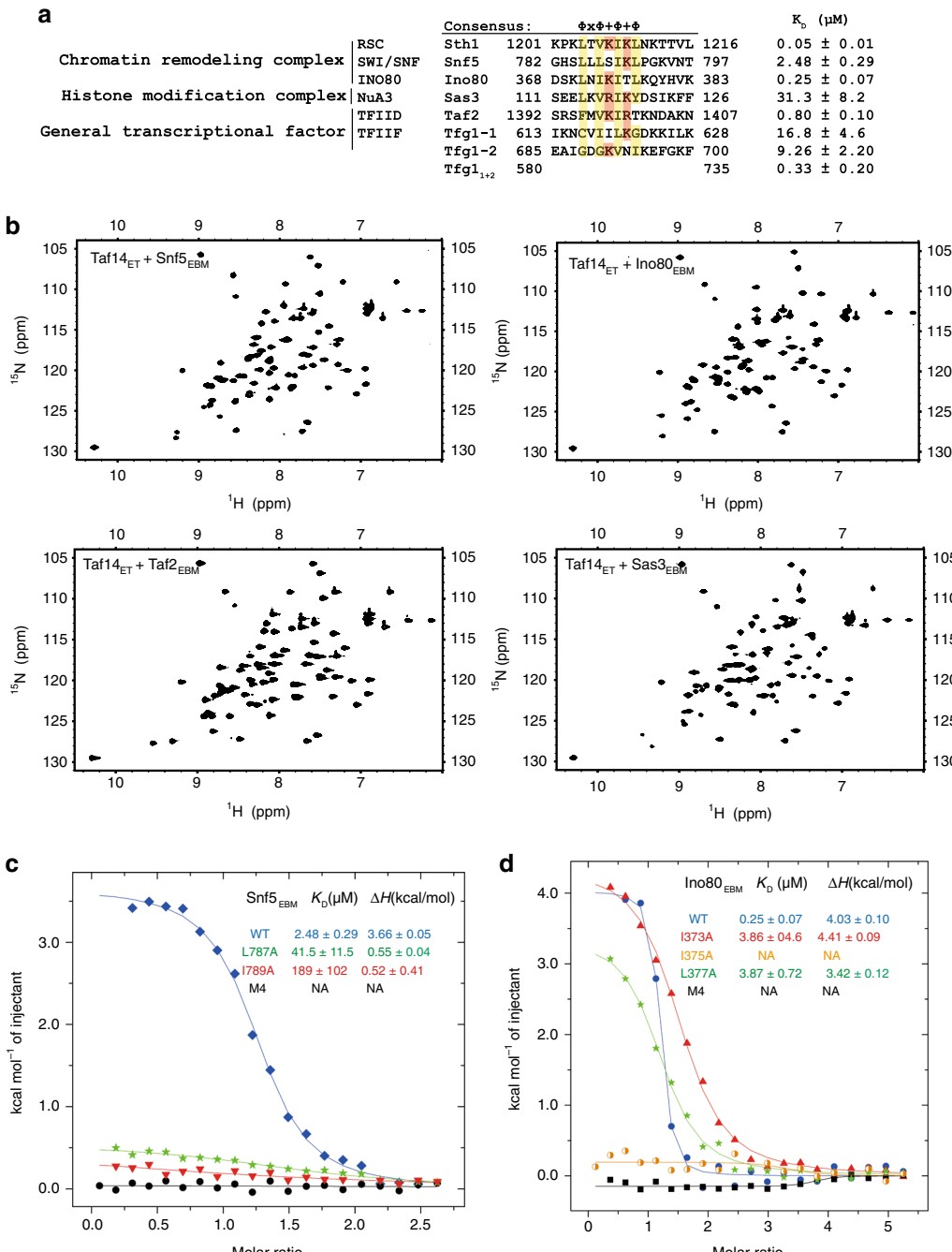

**Fig. 4 Taf14$_{ET}$ recognizes a conserved motif on partner proteins. a** Sequence alignments of the Taf14-binding motifs present in the Sth1, Snf5, Ino80, Taf2, Sas3, and Tfg1 proteins. The consensus sequence is $\Phi \times \Phi + \Phi + \Phi$, where $\Phi$ stands for hydrophobic residues, + stands for a positively charged residue, and × stands for any amino acid. The dissociation constants determined by ITC assays were shown. **b** [$^{15}$N-$^{1}$H] HSQC of Taf14$_{ET}$ (1 mM) in the presence of unlabeled Snf5$_{771-800}$, Ino80$_{368-390}$, Sas3$_{111-129}$, and Taf2$_{1393-1406}$ (1.5 mM). They all show similar chemical shift patterns that initially observed for the Taf14$_{ET}$–Sth1$_{EBM}$ complex. **c** ITC measurements reveal that mutations in hydrophobic residues of the Snf5 binding motif weaken the Taf14$_{ET}$–Snf5$_{EBM}$ interaction. M4 stands for the Snf5 L785A/L787A/I789A/L791A mutation. The dissociation constant ($K_D$), enthalpy change ($\Delta H$), and their fitting errors from one ITC plot were shown. **d** ITC measurements reveal that mutations in hydrophobic residues of the Ino80 binding motif weaken the Taf14$_{ET}$–Ino80$_{EBM}$ interaction. M4 stands for the Ino80 L371A/I373A/I375A/L377A mutation.

(compare Fig. 4b and Fig. 1d). Subsequent sequence alignment revealed a prominent consensus motif common to all of these Taf14$_{ET}$-binding short peptides: this motif comprises four hydrophobic residues that are interspersed by positive-charged residues (Fig. 4a). All these conserved residues contribute to the interfaces formed with Taf14$_{ET}$, as shown in the Taf14$_{ET}$–Sth1$_{EBM}$ complex structure. Indeed, we confirmed that mutations in these critical hydrophobic residues in Snf5$_{771-800}$ and Ino80$_{368-390}$

severely hindered the interaction with Taf14$_{ET}$ (Fig. 4c, d), further confirming the importance of hydrophobic contacts in Taf14$_{ET}$-mediated interactions.

This common sequence feature of Taf14$_{ET}$-binding motifs enabled us to swiftly identify the potential Taf14-binding motifs in Tfg1$_{580-735}$, which has previously reported to be a link between Taf14 and the TFIIF complex[24]. There are two motifs from Tfg1$_{580-735}$ containing a similar sequence pattern: one from 613

to 628, and the other one from 685 to 700 (Fig. 4a). We found that these motifs interacted with $Taf14_{ET}$ at $K_D$ values of around 17 and 9 μM, respectively; moreover, a fragment containing both motifs bound $Taf14_{ET}$ with a much higher binding affinity ($K_D$ around 0.33 μM; Supplementary Fig. 4f–i), indicating that a synergistic effect from multiple binding sites of Tfg1 facilitates its interaction with $Taf14_{ET}$. Taken together, these results clearly suggest that $Taf14_{ET}$ can be understood as a protein–protein interaction module that recognizes a consensus motif in multiple subunits of several nuclear complexes.

**Partner binding enhances phase separation of $Taf14_{ET}$.** After elucidating that $Taf14_{ET}$ is a versatile protein interaction domain, we further explored how $Taf14_{ET}$ coordinates its interactions with different binding partners. We hypothesized that $Taf14_{ET}$ might serve as a "hub" to orchestrate the actions of multiple complexes. Pursuing this, and given our results showing that $Taf14_{ET}$ is prone to form soluble aggregate at high concentration in the absence of its binding partners, we initially tested whether $Taf14_{ET}$ can undergo phase separation under physiological salt concentration (150 mM NaCl) conditions. A $Taf14_{ET}$-GFP fusion protein was highly soluble and did not form droplets in the absence of crowding agents (Fig. 5a). However, when $Taf14_{ET}$ was added to the droplet formation buffer that contained 10% PEG8000 to mimic the crowded environment of the nucleus, the $Taf14_{ET}$ solution turned cloudy. Subsequent fluorescence microscopy analysis revealed that GFP-positive spherical droplets underwent dynamic fusion events in the PEG8000-containing samples, while GFP alone did not form droplets up to 25 μM (Fig. 5a, b; Supplementary Fig. 5a). The droplet formation occurred in a $Taf14_{ET}$ concentration-dependent manner (Fig. 5a). Fluorescence recovery after photobleaching (FRAP) experiments showed that $Taf14_{ET}$ droplets could be partially recovered in minutes after photobleaching (Fig. 5c, d). These dynamic and reversible characteristics observed for the $Taf14_{ET}$ droplets suggest that $Taf14_{ET}$ can undergo liquid–liquid phase separation (LLPS).

Recalling our finding that binding with $Sth1_{EBM}$ increases the observed structural stability of $Taf14_{ET}$, we extended our droplet formation assays to investigate how binding with its partners may modulate phase separation behavior of $Taf14_{ET}$ proteins. We initially hypothesized that LLPS of $Taf14_{ET}$ might be suppressed by $Sth1_{EBM}$ binding. However, turbidity measurements demonstrated that the addition of $Sth1_{1183-1240}$ increased the turbidity of $Taf14_{ET}$ solutions in the presence of 10% PEG8000 (Fig. 5e). Fluorescence images showed that $Sth1_{1183-1240}$-RFP, but not $Sth1_{1183-1240}^{M4}$-RFP, substantially augmented the intensity of $Taf14_{ET}$-GFP phase separation, with images indicating that the RFP signal was incorporated into the $Taf14_{ET}$-containing GFP droplets (Fig. 5f, g). As controls, RFP or $Sth1_{1183-1240}$-RFP did not form any droplet in our assay condition (Supplementary Fig. 5b, c).

We further tested the LLPS-promoting impact of $Taf14_{ET}$ binding with another binding partner, Snf5. Fluorescence imaging revealed that $Snf5_{771-800}$-CFP also promoted LLPS of $Taf14_{ET}$, similar to $Sth1_{1183-1240}$-RFP (Fig. 5g). Notably, similar experiments with samples containing $Taf14_{ET}$–GFP, $Sth1_{1183-1240}$-RFP, and $Snf5_{771-800}$-CFP proteins revealed the formation of droplets positive for signals from each of the three fluorescence reporter domains (GFP, RFP, CFP) (Fig. 5h), implying that the phase separation of $Taf14_{ET}$ assists the incorporation of multiple binding partners into the same condensed unit.

**There are two subtypes of ET domains.** After we revealed the binding mechanism and LLPS ability of the Taf14 ET domain, we wanted to extend our studies to other ET domains. A Dali search

revealed that the structure of $Taf14_{ET}$ closely resembled the ET domains from AF9, BRD3, and BRD4. Even though these four proteins share only minor similarity at the primary sequence level (Fig. 6a), structural superimposition revealed remarkable structural similarity among each of their ET domains when in complex with their binding peptides (Fig. 6b). In all of these complex structures, the binding peptide forms a β-strand that pairs with a β-strand positioned between α2 and α3 of the ET domain. Further, the superimposition showed that all of the critical hydrophobic interactions are well conserved (Fig. 6c).

Despite this overall structure similarity, there are some differences between the ETs of the Bromo-associated family proteins (including $BRD3_{ET}$ and $BRD4_{ET}$) and the ETs of the YEATS-associated family proteins (including $Taf14_{ET}$ and $AF9_{ET}$). Our data support the definition of a distinct form of ET domain specific to YEATS-containing proteins, which we refer to as YEATS–ET (YET) family, analogous to the well-known BET family. ET domains from BET and YET-family members have a variable loop configuration between α2 and α3 (Fig. 6a, b). The ET domains from BET-family members have a short α helix defined as $α_{BET}$ positioned between α2 and α3, while YET members replace $α_{BET}$ with a β-strand ($β_{YET}$), which participates in the formation of a three-strand β-sheet with interacting peptides.

These differences likely help explain the lack of any reports about binding partners common to BET- and YET-family proteins. We, therefore, compared the binding ability of $Sth1_{EBM}$ with two YET proteins (Taf14 from yeast and AF9 from human) and two BET proteins (Bdf1 and Bdf2 from yeast). Clearly supporting the functional diversions of these distinct domains, GST pull-down assays showed that $Sth1_{EBM}$ interacted with $AF9_{ET}$ and $Taf14_{ET}$ but had no interaction with $Bdf1_{ET}$ and $Bdf2_{ET}$ (Fig. 6d). Moreover, droplet formation assays showed that ET domains from YET and BET also had different phase separation abilities. Specifically, $Taf14_{ET}$ and $AF9_{ET}$ (two YET-family members) can readily undergo LLPS in vitro in the presence of 10% PEG8000 (Fig. 6e), whereas $BRD4_{ET}$ (a BET-family member) had much weaker LLPS ability, forming much smaller droplets than $AF9_{ET}$ and $Taf14_{ET}$ (Fig. 6e). Collectively, we define a functionally and structurally distinct family of ET domains specific to YEATS proteins, which may plausibly explain the distinct functions of BET vs. YET proteins in the nuclei.

## Discussion

Taf14 is an abundant protein identified in many protein complexes, with many studies reporting enriched interactions with transcriptional machineries including general transcription factors, mediator complexes, chromatin remodeling complexes, and acetylation complexes[6–11]. Taf14 is not a core component of these complexes, and removal of Taf14 does not affect the integrity of these complexes but alters the global transcription[3,26]. For a long time, the roles of Taf14 in these different complexes have remained elusive. Efforts to deepen our understanding of the precise role(s) of Taf14 have been hindered by a lack of separation-of-function mutations to dissociate Taf14 from a given complex without affecting others. Our structural insights revealed here enabled the design of separation-of-function mutations to disrupt the Taf14–Sth1 interaction and to precisely dissect the roles of Taf14 in the RSC complex. We noted that full-length Taf14–Sth1 interaction was severely decreased, but not eliminated, by the $Sth1^{M4}$ mutation which completely disrupted the $Taf14_{ET}$–$Sth1_{EBM}$ interaction (Fig. 2d, e), indicating that there might be additional interaction sites between Taf14 and Sth1 or between Taf14 and other RSC components. Notwithstanding, the identified $Taf14_{ET}$-$Sth1_{EBM}$ interface is the primary binding site

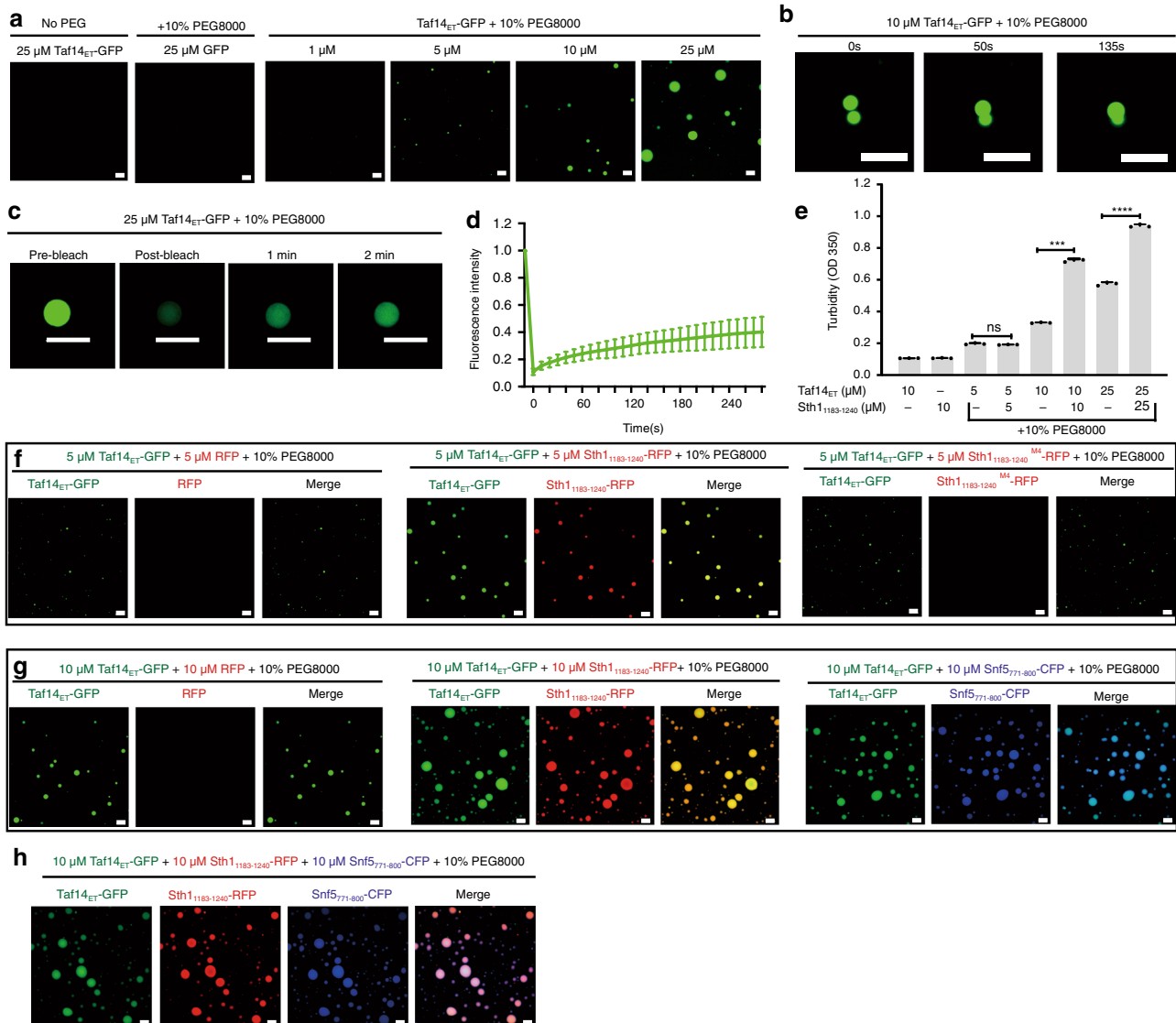

**Fig. 5 Phase separation of Taf14$_{ET}$. a** Representative fluorescence microscopy images of the Taf14$_{ET}$–GFP fusion protein at different concentrations. The droplet formation buffer comprises 25 mM Tris-HCl, pH 8.0, 150 mM NaCl, 10% PEG8000. The scale bars indicate 10 μm; note that the same scale applies to all of the panels. **b** Representative images of Taf14$_{ET}$-GFP at different time points showing the dynamic fusion of two droplets in the droplet formation buffer. **c** Representative images from FRAP experiments confirm the fluidity of Taf14$_{ET}$-droplets. **d** Quantification of the average intensity of recovered fluorescence in the bleached region of droplets at 10 s intervals. All of the data are presented as means ± SDs ($n = 16$). Source data are provided as a Source Data file. **e** Turbidity measurements as determined by light scattering at 350 nm. Sumo-Sth1$_{EBM}$ significantly increases the turbidity of GST-Taf14$_{ET}$ at 10 and 25 μM protein concentrations. Error bars represent standard deviations of three replicates. NS for $P > 0.05$; *** for $P < 0.001$; **** for $P < 0.0001$; two-tailed Student's $t$-test. Data are presented as mean ± SD, $n = 3$. **f** Representative images of the mixture of Taf14$_{ET}$-GFP (5 μM) and Sth1$_{EBM}$-RFP (5 μM) in the droplet formation buffer showing Sth1$_{EBM}$ increased the phase separation of Taf14$_{ET}$. As controls, RFP or Sth1$_{EBM}$$^{M4}$-RFP did not affect the phase separation of Taf14$_{ET}$. **g** Representative images of the mixture of Taf14$_{ET}$-GFP and Sth1$_{EBM}$-RFP or Snf5$_{EBM}$-CFP in the droplet formation buffer. Both Sth1$_{EBM}$ and Snf5$_{EBM}$ strongly promoted the phase separation of Taf14$_{ET}$ compared with the addition of RFP. **h** Representative images of a mixture containing the Taf14$_{ET}$-GFP, Sth1$_{EBM}$-RFP, and Snf5$_{EBM}$-CFP fusion proteins in droplet formation buffer. Note that all droplets contain all three fluorescence signals, indicating colocalization of three proteins.

between Taf14 and RSC, and disruption of this interface leads to the alteration of various pathways. Interestingly, the selective reduction of the Taf14-RSC association alters the transcription of genes involved in metabolism, which is consistent with the recent finding that Taf14 participates in yeast metabolic cycles[5]. It warrants further investigations to reveal what specific metabolic pathways are regulated by Taf14 and how the Taf14–Sth1 interaction guides the recruitment of RSC to specific chromatin loci to drive metabolic gene expression.

Besides the Taf14 association with RSC, we also revealed the structural basis of Taf14's association with the following complexes: SWI/SNF, INO80, TFIID, TFIIF, and NuA3 complexes. We have demonstrated how the mutations of Snf5 and Ino80 selectively disrupt their interactions with Taf14$_{ET}$ (Fig. 4c, d). Although whether Taf14 associates with these complexes solely through the identified Taf14$_{ET}$–EBM interfaces needs to be experimentally exploited, our study provides a toolbox to help decipher the unique roles of Taf14 with different nuclear complexes in future studies.

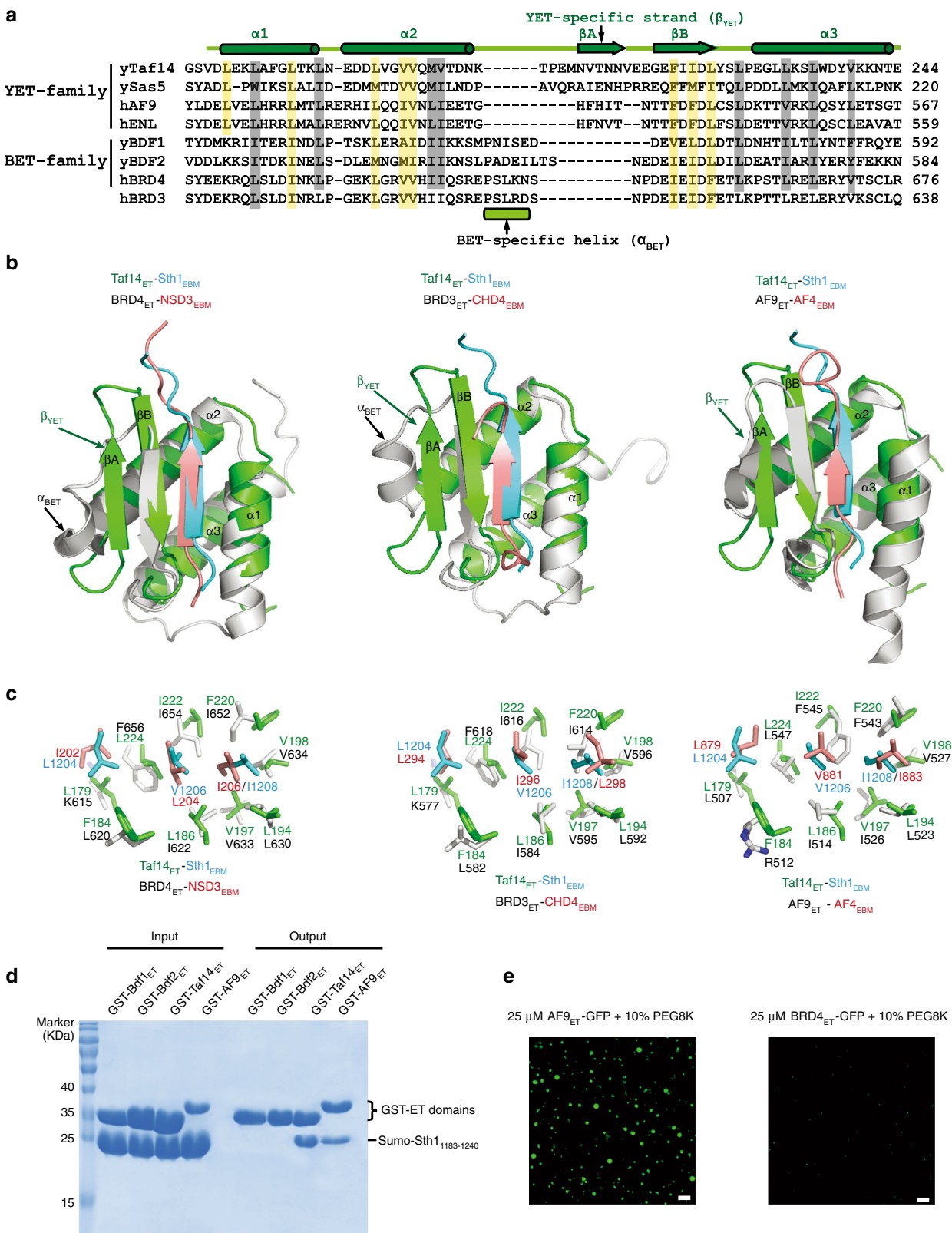

Here we propose one common function of Taf14 in different complexes. Taf14 may serve as the hub to bridge different transcriptional machineries at specific gene loci, which may explain its essential function in transcription regulation and non-essential role in the assembly of each associated complex (Fig. 7). Taf14 first binds to the acylated lysine markers (acetylation and crotonylation) on the chromatins targeted by its YEATS domain[3,12]. The abundant histone markers on chromatins can recruit a number of Taf14 molecules that concentrate on nucleosome fibers by the self-association ability of Taf14$_{ET}$ reported here. The phase-separated Taf14 molecules are capable of incorporating various transcriptional apparatus into one

**Fig. 6 Comparison of structural conserved YET and BET-family proteins. a** Sequence alignments of the ET domains from representative members of the YET family (yTaf14, ySas5, hAF9, and hENL) and the BET family (yBdf1, yBdf2, hBRD3, and hBRD4). Secondary structure assignments (based on the Taf14$_{ET}$ structure) are shown as cylinders (α-helices) and arrows (β-strands). The conserved hydrophobic residues critical for peptide binding are highlighted in yellow. The critical residues involved in the internal hydrophobic core formation are highlighted in gray. The YET-specific element (β$_{YET}$) and the BET-specific element (α$_{BET}$) are shown. **b** Structural comparison of the Taf14$_{ET}$–Sth1$_{EBM}$ complex with BRD4$_{ET}$-NSD3$_{EBM}$ (PDB: 2NCZ), with BRD3$_{ET}$-CHD4$_{EBM}$ (PDB: 6BGG), and with AF9$_{ET}$-AF4$_{EBM}$ (PDB: 2LM0). **c** Identification of the conserved hydrophobic interfaces common among these complex structures. The structurally equivalent residues are labeled. **d** GST pull-down assays revealed the differential binding interactions between Sumo-Sth1$_{1183-1240}$ and the various GST-tagged ET domains. **e** Representative images of the AF9$_{ET}$-GFP and BRD4$_{ET}$-GFP fusion proteins in the droplet formation buffer (25 mM Tris-HCl, pH 8.0, 150 mM NaCl, 10% PEG8000) at the final concentration indicated. The scale bars indicate 10 μm.

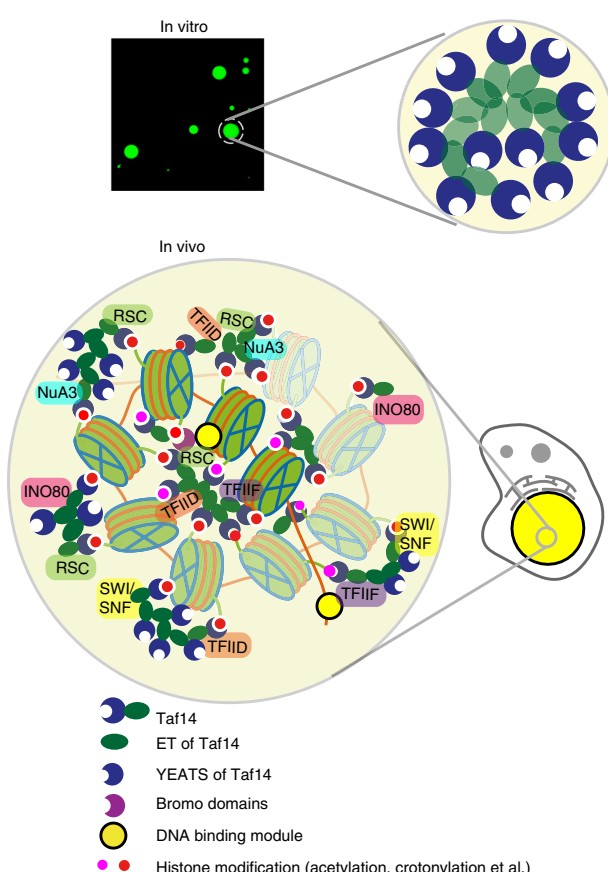

**Fig. 7 Proposed model of Taf14 as an organization hub to orchestrate transcription machinery recruitment.** Taf14$_{ET}$ alone and Taf14$_{ET}$-complexes undergo LLPS in vitro. Taf14 also contains an N-terminal YEATS domain recognizing acetylated and crotonylated histones. The existence of multiple Taf14$_{ET}$-binding motifs, multiple histone and DNA-binding modules in the complexes, as well as various modifications on chromatins can increase binding multivalence between Taf14-containing complexes and chromatins. These Taf14-mediated interactions consequently promote phase separation of condensates in which different transcriptional machinery is concentrated, thereby ensuring efficient transcription.

canonical intrinsic disordered region (IDR) or low complexity region (LCR), differing from other transcription factors and coactivators that use their IDR or LCR to mediate phase separation[28,29]. Thus, Taf14 is a unique example of coactivators that have both targeting and scaffolding roles through its structural domains.

Notably, some Taf14-binding partners (e.g. Tfg1) contain two or more ET-binding motifs, and different numbers of ET-binding sites may balance the stoichiometric ratio of different transcriptional complexes in the condensed transcriptional unit. The different affinities between Taf14 and binding partners may also elaborately regulate the dynamics of these complexes in the compartmentalized unit. It will be interesting to investigate how Taf14 serves as the central transcriptional hub in vivo to coordinate the actions of different complexes. Moreover, how post-translation modification or other factors could modulate the phase separation ability of Taf14 to regulate assembly or disassembly of the Taf14-centered transcriptional unit warrants further investigation.

Our structural and biochemical insights of Taf14 can be extended to a group of human transcription regulators containing both YEATS and ET domains (e.g., ENL and AF9), which we refer to as YET-family proteins. We show that the ET domain of AF9 can also form LLPS in vitro (Fig. 6e). A recent study reported that oncogenic mutations of ENL identified in Wilms tumor drove aberrant gene transcription by enhancing self-association ability of ENL, and the self-association ability of ENL mutants relied on the IDR region and the ET (also called AHD) domain of ENL[32]. Thus, it is a general feature for YET-family proteins to form the central hub to concentrate many complexes to a compartmentalized unit for efficient transcription, reinforcing the notion that YET-family proteins are master transcriptional regulators in cells[3,14–16]. We propose that targeting the ET domain by modulating its binding activity and phase separation ability might be a potential direction of therapeutic drug design for some ENL- and AF9-driven diseases.

It also bears emphasis that our study uncovered the similarity and difference between the well-known BET and the newly-defined YET-family proteins[14,33]. The variation in the binding pocket of ET domains leads to non-overlapped binding partners for these two families of proteins, suggesting that BET and YET are involved in independent functional pathways in transcriptional regulation. Moreover, the ET domain from the YET family (e.g., Taf14$_{ET}$ or AF9$_{ET}$) has a self-association ability to form phase-separated droplets in vitro, while the ET domain from the BET family (e.g., BRD4$_{ET}$) has a very weak LLPS ability (Fig. 6e). It should be noted that BRD4 could also form phase-separated condensates, but the phase separation of BRD4 is mediated by the C-terminal IDR region (residues 674–1351) instead of ET domain (residues 601–683)[29]. Thus, the common working mechanism for these two families of proteins is that they use N-terminal reader domains (YEATS or Bromo) to recognize the acylation markers on chromatins, and then use ET domains (and also IDR regions in BET and mammalian YET members) to concentrate required

condensing unit. Taf14-binding partners often contain additional histone-binding modules and DNA-binding modules, thus further boosting the binding multivalence between Taf14-containing complexes and chromatins, and consequently facilitating the formation of the condensates with concentrated transcriptional machineries to ensure efficient transcription (Fig. 7).

The role of Taf14 in scaffolding transcription machineries is consistent with the prevailing model that transcription factors and coactivators usually form high-concentration nuclear clusters by phase separation[27–29]. However, sequence analysis using PONDR[30] and SMART[31] show that Taf14 does not have the

complexes to the specific gene loci to efficiently initiate transcription, thus linking histone modification to transcription regulation.

## Methods

**Protein expression and purification.** *Saccharomyces cerevisiae* Taf14$_{FL}$ (residues 2–244), Taf14$_{ET}$ (residues 174–244), Sth1$_{EBM}$ (residues 1183–1240), and other fragments from Sth1, Snf5, Ino80, Sas3, Tfg1, and Taf2 subunit were amplified by high-fidelity FastPfu polymerase (Transgen Biotech, China), and then subcloned into a modified pET28b vector with an N-terminal 6*His-Sumo tag or a modified pGEX-6P-1 vector with 6*His-GST tag fused at the N terminus by ClonExpress II One Step Cloning Kit (Vazyme, China) (Supplementary Tables 2 and 3). The plasmids were extracted using SPARKeasy Superpure Mini Plasmid Kit (Sparkjade Science Co., Ltd., China) according to the manufacturer's instructions, and then transformed into *Escherichia coli* Rosetta cells. Proteins were overexpressed with induction by 0.2 mM Isopropyl β-D-1-thiogalactopyranoside (IPTG) for 16–18 h at 18 °C. Harvested cells were resuspended in the lysis buffer containing 50 mM Tris-HCl, pH 8.0, 400 mM NaCl, 10% glycerol, 2 mM 2-mercaptoethanol, and protease inhibitor cocktail (Bimake, China). The cells were broken by sonication on ice and then centrifuged at $20,000 \times g$ for 40 min. The supernatant was incubated with Ni-NTA beads (Qiagen, USA) for 2 h at 4 °C, and the tag-free proteins were then eluted by on-beads digestion with ULP1 protease (for pET28b-Sumo vectors) or 3C (for pGEX-6P-1 vectors) added at a molar ratio of 1:200. If the Sumo-fused or GST-fused proteins were required, the proteins were eluted by 300 mM imidazole or 15 mM reduced glutathione, respectively. Eluted proteins were further purified by Size-Exclusion Chromatography (SEC) on the Hiload Superdex 75 or Hiload Superdex 200 columns in the buffer of 20 mM phosphate buffer, pH 7.0, 100 mM NaCl, and 2 mM DTT. Purified proteins were concentrated and stored at −80 °C. All mutations were introduced by PCR-based site-directed mutagenesis, and the presence of appropriate mutations was confirmed by DNA sequencing. Mutant proteins were purified by using the same protocol as described above. Uniformly $^{15}$N- and $^{15}$N/$^{13}$C-labeled proteins were prepared from cells grown in the M9 minimal medium containing $^{15}$NH$_4$Cl with or without $^{13}$C$_6$-glucose and purified following the same protocol.

**GST pull-down assays.** For the pull-down of Sth1, Snf5, Ino80, Sas3, and Tfg1 by GST-tagged Taf14$_{ET}$, 25 μg of GST-tagged Taf14$_{ET}$ and 50 μg binding partners were mixed with 10 μL of glutathione–Sepharose 4B beads in 100 μL of binding buffer (25 mM Tris-HCl, pH 8.0, and 150 mM NaCl, 2 mM DTT). To increase the visibility of the short peptides on SDS-PAGE, all Snf5/Ino80/Sas3/Tfg1 fragments were purified as Sumo-fused proteins and used in the GST pull-down assays. 10 μL reaction mixtures were taken out as input controls. After incubation at 4 °C for 3 h, the beads were washed four times with 500 μL of the binding buffer. The bound samples were eluted with 25 μL elution buffer (15 mM reduced glutathione, 25 mM Tris-HCl, pH 8.0, 150 mM NaCl, 2 mM DTT). The input and eluted samples were analyzed with SDS-PAGE.

**Isothermal titration calorimetry (ITC).** For ITC measurements, peptides and Taf14$_{ET}$ proteins were dialyzed into the same assay buffer (20 mM phosphate buffer, pH 7.0, 100 mM NaCl, and 2 mM DTT). Protein concentration was measured based on its UV$_{280nm}$ absorption. ITC titrations were performed using a MicroCal ITC200 system (GE Healthcare) at 25 °C. Peptides at 1.0 mM in the syringe were titrated into Taf14$_{ET}$ at 0.1 mM in the sample cell. Each titration consisted of 17 successive injections (the first at 0.4 μL and the remaining 16 at 2.41 μL). The titration curves were processed using the Origin 7.0 software program (OriginLab) according to the "one binding site" fitting model. Each titration was repeated twice and one representative plot was shown in the paper. The dissociation constants ($K_D$), enthalpy changes ($\Delta H$), and the fitting errors were derived from one representative ITC plot.

**Circular dichroism.** Circular dichroism samples were diluted to 0.1–0.15 mg/mL with 2 mM HEPES, pH 7.0, 100 mM NaF buffer. All spectra were collected on a Chirascan v100 spectrometer (AppliedPhotophysics) from 180 to 260 nm with 0.5 s time-per-point at 25 °C. The data were analyzed using CDNN (Circlular Dichroism analysis using Neural Networks) software.

**Yeast strains.** The strains are in the BY4742 background (*MATα his3Δ1 leu2Δ0 lys2Δ0 ura3Δ0*). The plasmids for gene expression were constructed based on the pRS423 (Supplementary Table 1). These plasmids were transformed into yeast strains by a lithium acetate procedure. The plasmids carrying *TAF14* or *taf14$^{M2}$* were transformed into a *taf14Δ* strain (*MATα his3Δ1 leu2Δ0 lys2Δ0 ura3Δ0 taf14Δ:: KanMX4*). Since Sth1 is an essential gene and sth1Δ is lethal for yeast, we transformed diploid strain BY4743 (*MATa/α his3Δ1/his3Δ1 leu2Δ0/leu2Δ0 LYS2/ lys2Δ0 met15Δ0/MET15 ura3Δ0/ura3Δ0 sth1Δ:: KanMX4*) with pRS313 plasmids carrying either *STH1* or *sth1$^{M4}$* alleles. The transformants were sporulated and

subjected to tetrad dissection to isolate spore clones carrying *STH1* or *sth1$^{M4}$* plasmids for viability.

For CoIP assay, the *STH1* gene was tagged by a C-terminal TAP tag and *TAF14* gene was tagged by a C-terminal HA tag through regular homologous recombination. The *sth1$^{M4}$* was generated by CRISPR-Cas9 techniques. Approximately 1 μg of *sth1$^{M4}$* fragment with 100 bp homology arms and the pgRNA plasmid were co-transformed to the BY4742-2-Cas9 strain harboring pCas9. The positive colonies were verified by PCR sequencing. The pgRNA in the correct colony was removed by growing in 2% galactose and 3% raffinose instead of 2% glucose.

**Stress resistance assay and growth assay.** Yeast strains were cultured in YC medium at 30 °C and suspended in sterile water at a final concentration of OD$_{600}$ ~ 0.3. Five-fold serial dilutions were plated onto YC medium containing the different concentrations of NaCl, HU, MMS, and Phle. Plates were incubated at different temperatures for 2 days and photographed. Each assay was repeated three times, and all showed consistent results, so one representative result was shown. The yeast growth assays in the liquid medium were carried out in 5 mL YC medium with the starting OD$_{600}$ ~ 0.1. Cells were incubated at different temperatures for 48 h, and the growth curves were monitored by measuring OD$_{600}$ every 4 h.

**Co-immunoprecipitation.** Yeast cells from 1 L fresh cultures (OD 0.8–1.0) were harvested and washed with PBS buffer. The cells were suspended in 10 mL lysis buffer (100 mM HEPES-KOH pH 7.9, 100 mM potassium acetate, 10 mM magnesium acetate), and then broken by a beads-beater (Hualida, China). The lysates were centrifuged for 15 min at $23,700 \times g$ at 4 °C. The supernatant was mixed with 100 μL HA beads (Invitrogen) followed by incubation at 4 °C for 3 h. Beads were washed three times with 1 mL wash buffer (100 mM HEPES-KOH pH 7.9, 100 mM potassium acetate, 10 mM magnesium acetate, 2 mM EDTA, 0.1%NP-40), and the bound samples were eluted by 600 μL 1× SDS loading buffer. The samples were analyzed by western blot using TAP (Invitrogen, CAB1001, 1:1000) antibody, HA (Cell Signaling, 3724 s, 1:1000), and the HRP secondary antibody (Jackson ImmunoResearch, 111-035-003, 1:5000). The band intensity was quantified by ImageJ.

**Purification of RSC complex.** The C terminus of the Sth1 subunit of RSC was tagged with a TAP tag. The RSC complex was purified using a standard TAP procedure as described before[34]. Typically, 10 L of cells were harvested and broken by SPEX 6857 Freezer Mill (SPEXSamplePrep). The lysate was centrifuged at $200,000 \times g$ for 1 h at 4 °C. The supernatant was incubated with IgG-Sepharose beads (GE Healthcare) for 3 h at 4 °C. The protein sample was released from the beads by TEV protease digestion for 2 h at 4 °C. The sample was then incubated with calmodulin affinity beads (GE healthcare) for 3 h. The RSC complex was eluted in the elution buffer (25 mM Tris-HCl, pH 7.5, 150 mM NaCl, 20 mM EGTA, 2 mM DTT).

**RNA-seq analyses and qPCR.** RNA sequencing and analysis were performed by the BGI company (Beijing, China). Three samples of *STH1* strain and three samples of *sth1$^{M4}$* strain were sequenced. The average genome mapping ratio is 95.97%. Differentially expressed genes were identified using NOISeq[35] with fold change >2 and diverge probability >0.8. Gene ontology (GO) analyses were performed using DAVID (v6.7)[36]. GO categories were selected according to the $P$-values.

For qPCR, yeast cells (OD 0.8–1.0) were harvested and digested with zymolyase 20T (MP Biomedicals, LLC) to generate spheroplasts. Total RNA was extracted with TRIzol reagent (Life Technologies), and was reverse transcribed using TransScript First-Strand cDNA Synthesis SuperMix kit (TransGen Biotech, China). Diluted cDNAs were used as templates for qPCR with Fast SYBR™ Green Master Mix (Applied Biosystem) in an ABI ViiA7 thermocycler (Applied Biosystems). The relative expression level was derived by the standard curve methods, and all genes were normalized to the expression of ACT1. Statistical analyses were performed using a Student's *t*-test from four biological replicates.

**NMR spectroscopy.** Nuclear Magnetic Resonance (NMR) samples of the Taf14$_{ET}$ (residues 174–244) of 1 mM in complex with Sth1$_{EBM}$ (residues 1183–1240) of 1.5 mM were prepared in buffer containing 20 mM phosphate buffer, pH 7.0, 100 mM NaCl, 1 mM PMSF, 5 μL protease inhibitor cocktail, and 0.04% sodium azide in H$_2$O/D$_2$O (9/1) or D$_2$O. All NMR spectra were collected at 25 °C on Agilent DD2 600 MHz spectrometer equipped with a cryogenic probe or DD2 700 MHz spectrometer equipped with a HCN z-gradient room temperature probe. Chemical shifts were referenced to external DSS. Spectra were processed using the program NMRPipe (version 9.7)[37] and analyzed with the program SPARKY (version 3.115)[38]. The backbone assignments were obtained using HNCO, HNCACO, CBCA(CO)NH, HNCACB, HNCA, HN(CO)CA, HNHA, and $^{15}$N-edited NOESY-HSQC spectra. Aliphatic side-chain assignments relied on (H)CCH-TOCSY and H(C)CH-TOCOSY spectra[39]. Aromatic ring resonances were assigned using 3D $^{13}$C-edited NOESY spectra. A total of 113 of the 127 expected backbone peaks were

observed in the $^{15}$N-$^1$H HSQC recorded at 25 °C and pH 7.0. Ninety-two percent of the C and H resonances for all side chains have been assigned.

To reveal the NMR spectra of Taf14$_{ET}$ in complex with different peptides, these peptides were chemically synthesized by Scilight-Peptide and GenScript, China. The interaction of Taf14 with different peptides (Snf5$_{771–800}$, Ino80$_{368–390}$, Sas3$_{111–129}$, and Taf21$_{1393–1406}$) was monitored by collecting $^1$H-$^{15}$N HSQC spectra of the $^{15}$N-labeled Taf14$_{ET}$ (1 mM) at 25 °C. The data shown were obtained by using a 1:1.5 molar ratio of Taf14:peptides.

The backbone dynamics experiments were performed using an Agilent DD2 700 MHz spectrometer on the $^{15}$N-labeled sample[40]. The $^{15}$N-$^1$H NOE spectra were acquired with and without $^1$H saturation during the recycle delay (12 s) in an interleaved manner. The intensities of the NOE peaks were estimated with the SPARKY program, and the NOE ratios of the two states were calculated using the peak intensities in the presence and absence of proton saturation.

**NMR structure calculations.** Distance restraints for structure calculations were derived from cross-peaks in a simultaneous $^{15}$N and $^{13}$C-NOESY-HSQC ($\tau_m = 120$ ms) and $^{13}$C-edited aromatic NOESY-HSQC in $H_2O$ ($\tau_m = 120$ ms) respectively. NOE cross-peak assignment was obtained by using a combination of manual and automatic procedures. An initial fold of the protein was calculated on the basis of unambiguously assigned NOES, with subsequent refinement using the NOEassign module implemented in the program CYANA (version 2.1)[41]. Peak analysis of the NOESY spectra was generated by interactive peak picking with the program SPARKY. A total of 142 phi and psi torsion angle restraints were derived from the program TALOS+[42]. Hydrogen bond restraints were applied only for residues that were clearly in the secondary structure regions as judged by NOE patterns and chemical shifts and supported by TALOS+. The best 20 of 100 CYANA structures were subjected to molecular dynamics simulation in explicit water by the program CNS[43]. The final structures were inspected by PROCHECK and MolProbity using the PSVS software suite[44]. Structures were visualized using the program PyMOL 1.5 (http://pymol.sourceforge.net, Delano Scientific).

**Phase separation assays.** Purified Taf14$_{ET}$-GFP, Sth1$_{1183–1240}$-RFP and Snf5$_{771–800}$-CFP were dialyzed into the protein buffer (25 mM Tris-HCl, pH 8.0, 150 mM NaCl). The protein was then diluted to the desired concentration in the droplet formation buffer (25 mM Tris-HCl, pH 8.0, 150 mM NaCl, 10% PEG8000). To visualize the droplet formation, 3–6 μL of the protein solution was deposited onto a microscope slide. Fluorescent images were taken by a Zeiss LSM 710 microscope.

**Turbidity assay.** Phase separation of Taf14$_{ET}$-GST and Sth1$_{EBM}$-Sumo was induced as the droplet assay. Turbidity measurements were conducted by monitoring optical density at 350 nm (OD$_{350}$) in 96-well plates with 100 μL samples using SynergyNeo Multi-Mode Reader (Bio-Tek). All experiments were performed in triplicate.

**Fluorescence recovery after photobleaching (FRAP) analysis.** Fluorescence recovery after photobleaching (FRAP) experiments were performed on in vitro droplets using the 488 nm laser line of a 63 × 1.40NA Zeiss LSM 710. The droplets with 1–5 μm diameters were bleached for 100 iterations. After bleaching, the fluorescence intensities were measured and collected by Mean ROI (photobleached region) every 10 s. The raw data ($n = 16$) are processed and analyzed by GraphPad Prism.

**Statistics and reproducibility.** All GST pull-down assays and purification of RSC complex were performed at least twice. ITC experiments were repeated twice, and one representative plot was shown. The CoIP experiments were repeated twice, and the results from both experiments were shown in Fig. 2e and Supplementary Fig. 2e. The yeast stress resistance assays were repeated three times, and the representative results were shown. RNA-seq data were from three independent clones. The phase separation data were acquired from three times independent experiments, and more than 30 images were taken for each sample. They showed similar results, so the representative microscopy images were shown. Statistical analysis was performed using GraphPad Prism 8 software. The data are presented as mean ± SD. Two-tailed Student's $t$-test was used to compare the difference of two groups. * for $P < 0.05$, ** for $P < 0.01$, *** for $P < 0.001$ and **** for $P < 0.0001$.

**Reporting Summary.** Further information on experimental design is available in the Nature Research Reporting Summary linked to this paper.

## Data availability

The chemical shifts have been submitted to the BMRB (accession code 36309), and the structure ensemble and NOE restraint file have been deposited in the PDB with accession code 6LQZ. RNA-seq data are available in the SRA database under an accession code SRP238570. Other data are available from the corresponding author upon reasonable request. Source data are provided with this paper.

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

## Acknowledgements

We thank the staff members of the Large-scale Protein Preparation System, the Integrated Laser Microscopy System, and the Nuclear Magnetic Resonance System at the National Facility for Protein Science Shanghai (NFPS) for providing technical support. We thank Jinqiu Zhou for the yeast strains and vectors, Zhongjun Qin for yeast CRISPR-Cas9 plasmids. This work was supported by grants from the Strategic Priority Research Program of the Chinese Academy of Sciences (XDB37010303 to Y.C.), the National Natural Science Foundation of China (31670748 and 31970576 to Y.C.), and the National Natural Science Foundation of China (31661143021 to S.Q.).

## Author contributions

Y.C. conceived and supervised the project. G.C., D.W., F.Y., and Q.W. purified the proteins and performed biochemical assays. G.C., B.W., and H.X. performed NMR analyses. D.W. carried out yeast genetic manipulation and phenotype analyses. G.C., D.W., B.W., S.Q., and Y. C. prepared the figures and wrote the manuscript.

## Competing interests

The authors declare no competing interests.
