## [Peer Review File · Nature Communications]

Reviewers' comments:

Reviewer #1 (Remarks to the Author):

In this manuscript, Chen and coworkers provide a structural and functional characterization of the Taf14 extra-terminal (ET) domain in isolation and in complex with various coactivator proteins. From the structural and functional studies, the authors conclude that disorder and the potential for multivalent interactions of Taf14 enables Taf14 to function as a molecular hub to coordinate transcription. Furthermore, the authors report that Taf14 is able to undergo liquid-liquid phase separation alone and in complex with its binding partners. This type of behavior has been observed in a multitude of other systems involving disordered proteins and transcriptional regulators and is not quite surprising, especially with the use of crowding agents as employed here. However, the observation that Taf14 ET undergoes a co-folding process upon binding to other disordered peptide sequences containing a common motif is very interesting and expands our current views of how transcriptional regulation is achieved via hub proteins. The data are presented clearly and convincingly, and this manuscript should be of broad interest to the readers of Nature Communications. I suggest some additional experiments and minor comments below that should improve the quality of the manuscript, address some technical concerns, and bolster the authors' claims of intrinsic disorder in the Taf14 ET domain.

1. There is a potential alternate explanation for the appearance of the NMR spectrum of the TAF14 ET domain in isolation (Fig. 1D). It appears that far fewer resonances are observed than would be expected based on the sequence of the construct used for this experiment. This spectrum could also represent a soluble aggregate, where the only resonances that are observable are those that remain dynamic and are not subject to size-induced broadening. What concentration of ¹⁵N-labeled Taf14 ET was used to obtain the spectrum shown? To probe potential self-association of Taf14 ET (which would be consistent with the microscopy data), the authors could collect NMR spectra at a range of concentrations, to observe whether the spectra improve (more peaks, better resolution) at lower concentrations. Additionally, CD spectra of Taf14 ET should be obtained for Taf14 ET in isolation and in complex with its binding partners to determine whether Taf14 ET is indeed disordered in its unbound state, or whether it is sampling a more molten globular/semi-structured state in solution.

2. A point of confusion—different K_d values are given for Sth1 in Fig. 1C and Fig. 4A. Were these values obtained for the same construct of Sth1? For purposes of comparison with the other binding partners, for which minimal constructs were used, it would be helpful to have a comparison to a peptide containing the minimal binding motif of Sth1 (ie. the ordered segment identified from the NMR structure). It is important to make consistent comparisons between the various binding partners, and despite the disordered nature of the flanking regions in the Sth1 construct, these regions have been shown in many recent papers to have an influence on binding affinity complex stability.

Other general comments:

- Error bars are largely missing in the presentation of the experimental data. For ITC experiments, how are the uncertainties in K_d values estimated? Why are there no uncertainties specified for the thermodynamic parameters?
- Thermodynamic parameters should be represented in kJ/mol.
- Pay attention to significant figures.

Reviewer #2 (Remarks to the Author):

Review of "The extra-terminal domain of Taf14 recognizes a common motif in diverse transcriptional machineries and facilitates their clustering by phase separation"

In this manuscript, Chen et al. interrogate the mechanism by which the YEATS containing, chromatin binding protein Taf14 coordinates interactions between itself and the up to 7 distinct complexes with which it interacts in the *S. cerevisiae* model organism. Taf14, along with its metazoan ortholog Af9, interact with acylated histone proteins to facilitate a gene regulatory response. Through structural biology approaches and detailed biophysical measurements, the authors identify a common peptide motif present in subunits of each of six distinct transcription-related complexes to which Taf14 binds. As a test case, the authors make a separation of function mutant in Sth1, a component of the RSC chromatin remodeling complex, to reduce binding between Sth1 and Taf14 without disrupting the RSC complex. This mutant displays both growth defects as well as alterations in gene regulation. Finally, the authors demonstrate that Taf14, under protein crowding conditions, can form phase separated liquid droplets. Furthermore, these droplets can partition peptides from distinct transcription complexes into the same phase separated liquid droplet. This observation led the authors to propose a mechanism for Taf14 function: through its YEATS domain acylated histone binding activity, Taf14 partitions to specific genomic locations to regulate transcription; and through phase separation and its ability to bind multiple transcription-related complexes, Taf14 coordinates the activities of these 7 different transcription-related complexes in a phase separated and concentrated location, similar to mechanisms observed in metazoan super-enhancers. The data presented in the submitted manuscript are compelling and the interpretations described by the authors are consistent with the data. In addition, the study is of interest to the broader transcription regulation field and timely in its findings. Therefore, with minor revisions addressed below, it is the opinion of this reviewer that the manuscript be accepted for publication:

1. Either the methods section or the figure legends should include more information regarding experimental reproducibility. The number of times each experiment was performed should be included in these descriptions. Specifically, the number of biological replicates used for growth assays and the number of times the coIP experiments were performed.
2. Either commentary or additional experiments should be provided to demonstrate that interactions between Taf14 and the intact protein complexes are abrogated upon mutation of the extra terminal binding domains instead of just relying on peptide interaction assays. In the ITC experiments, binding between the Sth1 peptide and Taf14 is completely abrogated in the m4 mutant. However, the coIP experiment only shows a reduction of Taf14 binding, not an elimination. While the coIP data is supported by growth data, qPCR and RNAseq analyses, it could be bolstered with quantitation of the coIP analyses including multiple replicates. Furthermore, the authors specifically mention that Tfg1 contains two distinct sequences that can interact with Taf14. Taf2 was shown to have multiple domains that can interact with Taf14 in a cited paper. In the full length Taf2 in the context of TFIID, mutation of either of the domains was not sufficient to abrogate binding between Taf14 and TFIID. While the identified peptide sequence in the EBD obviously contributes to Taf14 binding in the peptides tested, without testing of the intact protein, it is not possible to show that this peptide is essential for interaction for each of the different complexes. Considering this type of analysis was performed for Sth1 and RSC, the absence of these experiments should not preclude publication.
3. In line 205, when describing the temperature of the heat sensitivity, the authors claim that the cells grew normally below 32°C. However, this temperature was not tested, as shown in figure 3. This data point should be included in the figure or the text should be amended to "below 30°C". In addition, the sth1m4 mutant appears to have a modestly slower growth phenotype compared to the WT even at lower temperatures. Growth curves, while not necessary for this study, could provide a more quantitative reading of growth rates comparing the two strains.
4. The phase separated liquid droplet data would be even more compelling if the mutant peptides were also included in the analyses. This would demonstrate that the partitioning of the peptides to the Taf14 droplets is dependent on the amino acids required for nanomolar binding affinity.

Reviewer #3 (Remarks to the Author):

Chen et al. report that the ET domain of TAF-14 binds a common motif (EBM) in unstructured sequences of several nuclear factors, as judged by GST pulldown assays. They solve the solution structure of the Taf14 ET domain with the EBM of Sth1. Based on the structure, they design mutants disrupting this interaction and analyse their affinity by ITC. They further find similar interaction patterns for several other EBM peptides by ITC.

Next, they use a quadruple Sth1 mutant that completely abolishes interaction with Taf14 in ITC. They find that in yeast cells, if this mutant is the only copy expressed, it strongly diminishes interaction with Taf14, has a growth defect at higher temperatures and on alternative carbon sources, and changes the expression of about 400 genes.

Last, they demonstrate that the Taf14-ET can undergo phase separation, and that phase separation is stimulated by EBM peptides.

They demonstrate that EBM binding and phase separation is conserved for YEATS but not Bromo associated ET domains, across species.

The experiments are mostly very clean and logical, and I would in principle recommend publication.

However, I have a couple of technical and stylistic comments

Major points:

(1) I was wondering about the structural integrity of the Taf14 mutants in the ITC experiments (Figure 2D, line 172 etc), arginine might be quite disruptive here. ^[1] Could the authors present some kind of data supporting addressing this point, for example gel filtration profiles or NRM, or alternatively use something smaller (Alanine or Serine) or maybe find a 'rescuing' mutant on the Sth1 side?

(2) For the in vivo experiments, the authors work with the Sth1-M4 mutation, and claim that this disrupts specifically the Taf14 interaction. However, this could also affect other proteins (ET domains?) binding to the same motif. ^[2] Also, could the authors please quantify the WB in Figure 2E? It looks as if the M4 mutant was already a bit less in the input.

(3) To be better able to judge the quality of the data, it would be essential to show the original ITC curves for the experiments in Figure 4A, not only the final Kd values; only some of them are shown in Figure 4C/D

(4) For the phase separation (LLPS) assays, the authors should provide more controls. ^[3] My first concern is the Taf14 protein they use - does it still carry the GST tag, and what GFP variant are they using? 'Canonical' GFP tends to dimerise, and can undergo phase separation by itself, especially in strongly crowding conditions like 10% PEG. ^[4] Second, I am wondering about the addition of the EBM factors; this could also be additional protein crowding. As a control, they should use RFP only (and whatever else is attached to these peptides)

Minor points:

(1) In the introduction, it might be nice to introduce Sth1 with a sentence or two, since it will play a major role in this paper

(2) In Supplementary Figure 2A, is it possible that the labelling of the ITC curves (for I1208A and L1210A) was mixed up?

(3) For gene expression analysis, the authors use RNA sequencing and see differences when comparing the transcriptome of the wild-type versus the Sth1-M4 carrying strain. The experiments performed seem valid to me. I am aware that this might go beyond the scope of this manuscript, but the ultimate proof of their model would be to demonstrate that Taf14 or RSC recruitment is affected at the genes where they see changes.

- (4) See major comment 4: a list of plasmids / expression constructs would be useful
- (5) I find the structure in Figure 2C a bit hard to read, especially which amino acids interact. Maybe they can find a clearer perspective, or show two alternative ones ?
- (6) In Figure 6E, a scale bar would be helpful.
- (7) At some selected points, the English grammar was not yet perfect, it might be good to send it to a native speaker for a quick read-through.

Reviewer #4 (Remarks to the Author):

This manuscript reported that the extra-terminal domain of Taf14 recognizes a common motif in multiple transcriptional coactivator proteins from diverse nuclear complexes, and the authors demonstrate that Taf14ET could form liquid-liquid phase separation (LLPS) in vitro. This manuscript not only identifying the molecular mechanism of Taf14ET-related transcriptional mechanisms, but helps us to understand the diverse cellular processes that Taf14ET regulating. However, the functions of Taf14-Sth1 interaction have not fully studied. I believe it is important for this manuscript, this is also why we investigate the transcriptional machineries of Taf14. So, some improvements might need to take into consideration before publication. In this case, the decision of this manuscript is major revision.

Some comments and suggestions:

Major comments:

1. Why the authors use 'Sth1' as an example? I suggest the author explain the reason in Introduction section.
2. Line 116: In figure 1B, the authors mentioned 'GST-Taf14¹⁷⁴⁻²⁴⁴ could pull-down Sth1¹¹⁸³⁻¹³⁵⁹, Sth1¹¹⁸³⁻¹²⁴⁰, and 587 Sth1¹¹⁹⁹⁻¹²²⁵, but not Sth1¹²⁴⁸⁻¹³⁵⁹.' But in Line 116, the author reported that 'the Sth1 fragments comprising residues 1183-1240 and 1199-1225 were able to associate with Taf14ET'. Why the Sth1¹¹⁸³⁻¹³⁵⁹ is lost?
3. In Line 120, the authors mentioned that 'the interaction between Taf14ET and Sth1¹¹⁸³⁻¹²⁴⁰ is dominated by hydrophobic interactions rather than electrostatic contacts', but I do not know why the authors indicated that 'Electrostatic interactions further stabilize the Sth1^{EBMC}-Taf14ET interaction.'? So, the interaction was dominated by hydrophobic and electrostatic interactions?
4. Line 139: If we deleted the 9-residue core fragment, and determined the binding capacity? If so, we can further confirm that this motif directly interacts with Taf14ET through mutagenesis study.
5. Why the authors choose temperatures as the phenotype's experiments? I suggest the authors choose different types of stress conditions to confirm the universality of the conclusion. Such as various pH, osmotic pressure. In Figure 3A, the authors did not study the phenotype at 32 °C, how to ensure that the strain grew at 32 °C and below? Figure 3B, how to determine the appropriate concentration of the DNA-damage stress conditions? It is difficult to display that no sensitivity to MMS was detected in figure 3B. The author is advised to adjust the contrast of the figure or find a suitable dilution gradient.
6. Line 219: Did the authors select 7 genes for verification? If so, based on what? In addition, the expression levels of these genes in the RNA-seq experiments should also be marked in Figure 3D.
7. Line 227: I do not think the authors have well studied the metabolic pathways that Taf14-Sth1 interaction participated.
8. Line 239: where are the results of Taf2? I have not found the corresponding figure.

Minor comments:

1. Line 63: 'Saccharomyces cerevisiae' changed to 'S. cerevisiae', please check other expression.
2. Line 93: as an example
3. Line 98: 'analysis' to 'analyses'.
4. Line 119: dissociation constant (K_d); '55 nM'
5. Line 125: 'nuclear magnetic resonance (NMR)' Abbreviations should be marked when they first appear. In addition, the author needs to check other abbreviations. Some abbreviations need to be marked with their full names.

6. Line 201: 'sth1'
7. Line 227: there is no figure 3H, is it figure 3F?
8. Line 261: In Supplementary 4D, is there any control group?
9. Line 287: 'Taf14ETs'
10. Line 336: Please marked the input and output in figure 6D.

Reviewer #1 (Remarks to the Author):

In this manuscript, Chen and coworkers provide a structural and functional characterization of the Taf14 extra-terminal (ET) domain in isolation and in complex with various coactivator proteins. From the structural and functional studies, the authors conclude that disorder and the potential for multivalent interactions of Taf14 enables Taf14 to function as a molecular hub to coordinate transcription. Furthermore, the authors report that Taf14 is able to undergo liquid-liquid phase separation alone and in complex with its binding partners. This type of behavior has been observed in a multitude of other systems involving disordered proteins and transcriptional regulators and is not quite surprising, especially with the use of crowding agents as employed here. However, the observation that Taf14 ET undergoes a co-folding process upon binding to other disordered peptide sequences containing a common motif is very interesting and expands our current views of how transcriptional regulation is achieved via hub proteins. The data are presented clearly and convincingly, and this manuscript should be of broad interest to the readers of Nature Communications. I suggest some additional experiments and minor comments below that should improve the quality of the manuscript, address some technical concerns, and bolster the authors' claims of intrinsic disorder in the Taf14 ET domain.

We thank this reviewer's appreciation of our work.

1. There is a potential alternate explanation for the appearance of the NMR spectrum of the TAF14 ET domain in isolation (Fig. 1D). It appears that far fewer resonances are observed than would be expected based on the sequence of the construct used for this experiment. This spectrum could also represent a soluble aggregate, where the only resonances that are observable are those that remain dynamic and are not subject to size-induced broadening. What concentration of ¹⁵N-labeled Taf14 ET was used to obtain the spectrum shown? To probe potential self-association of Taf14 ET (which would be consistent with the microscopy data), the authors could collect NMR spectra at a range of concentrations, to observe whether the spectra improve (more peaks, better resolution) at lower concentrations. Additionally, CD spectra of Taf14 ET should be obtained for Taf14 ET in isolation and in complex with its binding partners to determine whether Taf14 ET

is indeed disordered in its unbound state, or whether it is sampling a more molten globular/semi-structured state in solution.

We thank this reviewer's insightful suggestion. The reviewer is right, and we made a wrong conclusion in the original submission.

We initially collected NMR spectra of apo Taf14_{ET} at 0.5 mM and 1 mM concentration, and both showed far fewer resonances than expected. According to the reviewer's suggestion, we re-collected NMR spectra of apo Taf14_{ET} at different lower concentrations (0.05, 0.1, 0.2, and 0.5 mM) (Figure 1d). To our surprise, the peaks of 0.05 mM Taf14_{ET} were well-dispersed with some overlapping signals in the 8-8.5 ppm range. When the Taf14_{ET} concentration was increased to 0.2 mM, the majority of HSQC peaks were poorly dispersed with the intense overlapping signals within a narrow 7.5-8.5 ppm chemical shift range, similar as what we previously observed in 0.5 mM Taf14_{ET} sample (Figure 1d). It strongly suggests that the soluble aggregation, but not the unfolded structure, results in the poor-dispersed NMR spectra of apo Taf14_{ET}. We also used circular dichroism (CD) to characterize the structures of Taf14_{ET}, Sth1_{EBM} and the Taf14_{ET}-Sth1_{EBM} complex. Both Taf14_{ET} alone and in complex with Sth1_{EBM} peptide showed similar spectra features representing a mixed α - β structure (Figure 1f). These results indicate that Sth1_{EBM}-binding stabilizes Taf14_{ET} conformation in solution but not inducing dramatic conformational change.

The manuscript has been revised accordingly, and new data are shown in Fig. 1d (Taf14_{ET} HSQC at different concentration) and Fig. 1f (CD analyses).

2. A point of confusion—different K_d values are given for Sth1 in Fig. 1C and Fig. 4A. Were these values obtained for the same construct of Sth1? For purposes of comparison with the other binding partners, for which minimal constructs were used, it would be helpful to have a comparison to a peptide containing the minimal binding motif of Sth1 (ie. the ordered segment identified from the NMR structure). It is important to make consistent comparisons between the various binding partners, and despite the disordered

nature of the flanking regions in the Sth1 construct, these regions have been shown in many recent papers to have an influence on binding affinity complex stability.

We are sorry for the confusion. The ITC values in Fig. 1c and Fig. 4a are from two batches of ITC experiments using the same Taf14 and Sth1 constructs. These two ITC experiments yielded similar but not identical K_D values (55 nM and 67 nM). To avoid confusion, we have replaced the K_D value in Fig. 4a with the K_d value from Fig. 1c.

For ITC assays, we used the Sth1 construct 1183-1240, which is much longer than the core region (1203-1211) seen in the NMR structure. We agree with the reviewer's concern that the flanking sequence around the core binding region could affect the complex stability. We have tried to purify several short Sth1 peptides covering the core segment as the reviewer suggested (Sth1₁₂₀₁₋₁₂₁₂ and Sth1₁₁₉₉₋₁₂₁₇). However, these peptides were too hydrophobic to be soluble in the regular ITC assay buffer. Thus, we cannot characterize the interaction between these minimal Sth1 fragments and Taf14_{ET}. On the other hand, the data in Fig. 4a is not intended to compare the binding affinities of different binding partners. The primary purpose of Fig. 4a is to demonstrate that these fragments can interact with Taf14_{ET} and have a similar sequence pattern.

Other general comments:

- Error bars are largely missing in the presentation of the experimental data. For ITC experiments, how are the uncertainties in K_d values estimated? Why are there no uncertainties specified for the thermodynamic parameters?

For ITC experiments, the plot is from a single experiment and the uncertainties is derived from the fitting error of the single ITC plot. Now we have indicated the source of the error in the method section. We also included the fitting error for thermodynamic parameters.

- Thermodynamic parameters should be represented in kJ/mol.

Thanks. Now the ΔH is shown in kcal/mol.

- Pay attention to significant figures.

We have made some revision on the figures to make them clearer to readers.

Reviewer #2 (Remarks to the Author):

Review of “The extra-terminal domain of Taf14 recognizes a common motif in diverse transcriptional machineries and facilitates their clustering by phase separation”

In this manuscript, Chen et al. interrogate the mechanism by which the YEATS containing, chromatin binding protein Taf14 coordinates interactions between itself and the up to 7 distinct complexes with which it interacts in the *S. cerevisiae* model organism. Taf14, along with its metazoan ortholog Af9, interact with acylated histone proteins to facilitate a gene regulatory response. Through structural biology approaches and detailed biophysical measurements, the authors identify a common peptide motif present in subunits of each of six distinct transcription-related complexes to which Taf14 binds. As a test case, the authors make a separation of function mutant in Sth1, a component of the RSC chromatin remodeling complex, to reduce binding between Sth1 and Taf14 without disrupting the RSC complex. This mutant displays both growth defects as well as alterations in gene regulation. Finally, the authors demonstrate that Taf14, under protein crowding conditions, can form phase separated liquid droplets. Furthermore, these droplets can partition peptides from distinct transcription complexes into the same phase separated liquid droplet. This observation led the authors to propose a mechanism for Taf14 function: through its YEATS domain acylated histone binding activity, Taf14 partitions to specific genomic locations to regulate transcription; and through phase separation and its ability to bind multiple transcription-related complexes, Taf14 coordinates the activities of these 7 different transcription-related complexes in a phase separated and concentrated location, similar to mechanisms observed in metazoan super-enhancers. The data presented in the submitted manuscript are compelling and the interpretations described by the authors are consistent with the data. In addition, the study is of interest to the broader transcription regulation field and timely in its findings. Therefore, with minor revisions addressed below, it is the opinion of this reviewer that the manuscript be accepted for publication.

We thank this reviewer's appreciation of our work.

1. Either the methods section or the figure legends should include more information regarding experimental reproducibility. The number of times each experiment was performed should be included in these descriptions. Specifically, the number of biological replicates used for growth assays and the number of times the coIP experiments were performed.

We thank the reviewer for this suggestion. In the revision, we have included more information about experimental reproducibility in both methods section and figure legends. To directly answer the reviewer's concerns, the growth assays were replicated three times, and all generated similar phenotypes. The Co-IP experiments were repeated twice and both showed similar results. One CoIP result is now presented in the main figure 2e and the other one is now presented in the supplementary figure 2e.

2. Either commentary or additional experiments should be provided to demonstrate that interactions between Taf14 and the intact protein complexes are abrogated upon mutation of the extra terminal binding domains instead of just relying on peptide interaction assays. In the ITC experiments, binding between the Sth1 peptide and Taf14 is completely abrogated in the m4 mutant. However, the coIP experiment only shows a reduction of Taf14 binding, not an elimination. While the coIP data is supported by growth data, qPCR and RNAseq analyses, it could be bolstered with quantitation of the coIP analyses including multiple replicates. Furthermore, the authors specifically mention that Tfg1 contains two distinct sequences that can interact with Taf14. Taf2 was shown to have multiple domains that can interact with Taf14 in a cited paper. In the full length Taf2 in the context of TFIID, mutation of either of the domains was not sufficient to abrogate binding between Taf14 and TFIID. While the identified peptide sequence in the EBD obviously contributes to Taf14 binding in the peptides tested, without testing of the intact protein, it is not possible to show that this peptide is essential for interaction for each of the different complexes. Considering this type of analysis was performed for Sth1 and RSC, the absence of these experiments should not preclude publication.

We thank the reviewer for raising this point. In the revision, we have provided the quantification data of WB bands in CoIP assays (Fig. 2e). The CoIP using full-length Taf14 and Sth1 indeed only showed the severe reduction (~70% reduction) of the Sth1-Taf14 interaction for Sth1^{M4} mutant, but the ITC assays using protein fragments showed the complete disruption of the Sth1-Taf14 interaction. It suggests that, although our identified Taf14_{ET}-Sth1_{EBM} interface is the primary binding site between these two proteins, there might be additional interaction sites between Taf14 and Sth1 or between Taf14 and other RSC components. We have discussed this possibility in the discussion section.

We agree with the reviewer that further studies are required to elucidate whether Taf14 associates with any intact complex through the proposed short motifs. It will be not unexpected that additional elements are also needed for Taf14 association with different complexes. We have discussed this point in the discussion section.

3. In line 205, when describing the temperature of the heat sensitivity, the authors claim that the cells grew normally below 32oC. However, this temperature was not tested, as shown in figure 3. This data point should be included in the figure or the text should be amended to “below 30oC”. (Sorry, it should be 30C). In addition, the *sth1m4* mutant appears to have a modestly slower growth phenotype compared to the WT even at lower temperatures. Growth curves, while not necessary for this study, could provide a more quantitative reading of growth rates comparing the two strains.

We thank the reviewer for pointing out the typo. It should be “the cells grew normally below 30°C”, not 32°C. According to the reviewer’s suggestion, we have provided growth curves to directly show the growth defects caused by *sth1 M4* mutation (Supplementary Figure 3d). The *sth1^{M4}* mutant showed much-delayed growth when the temperature was above 30°C, but had similar (albeit slightly slow) growth rate compared to the WT strains.

4. The phase separated liquid droplet data would be even more compelling if the mutant peptides were also included in the analyses. This would demonstrate that the partitioning of the peptides to the Taf14 droplets is dependent on the amino acids required for nanomolar binding affinity.

Thanks for this suggestion. In the revision, we have included the phase separation data for the mutant Sth1 (Sth1 M4) peptide (Figure 5f). It clearly showed that Sth1 M4 peptide lost the ability to promote the phase separation of Taf14_{ET}.

Reviewer #3 (Remarks to the Author):

Chen et al. report that the ET domain of TAF-14 binds a common motif (EBM) in unstructured sequences of several nuclear factors, as judged by GST pulldown assays. They solve the solution structure of the Taf14 ET domain with the EBM of Sth1. Based on the structure, they design mutants disrupting this interaction and analyze their affinity by ITC. They further find similar interaction patterns for several other EBM peptides by ITC.

Next, they use a quadruple Sth1 mutant that completely abolishes interaction with Taf14 in ITC. They find that in yeast cells, if this mutant is the only copy expressed, it strongly diminishes interaction with Taf14, has a growth defect at higher temperatures and on alternative carbon sources, and changes the expression of about 400 genes.

Last, they demonstrate that the Taf14-ET can undergo phase separation, and that phase separation is stimulated by EBM peptides. They demonstrate that EBM binding and phase separation is conserved for YEATS but not Bromo associated ET domains, across species.

The experiments are mostly very clean and logical, and I would in principle recommend publication.

We thank this reviewer's appreciation of our work.

However, I have a couple of technical and stylistic comments.

Majorpoints:

(1) I was wondering about the structural integrity of the Taf14 mutants in the ITC experiments (Figure 2D, line 172 etc), arginine might be quite disruptive here. Could the authors present some kind of data supporting addressing this point, for example gel filtration profiles or NMR, or alternatively use something smaller (Alanine or Serine) or maybe find a 'rescuing' mutant on the Sth1 side?

We thank the reviewer for raising this issue. Now we have included gel-filtration profiles of Taf14 F220R, I222R and WT in the Supplementary Figure 2d. It showed that these arginine mutations did not affect the overall structural integrity of Taf14_{ET}. Moreover, we included one more mutation F220A, as the reviewer suggested. Taf14 F220A also decreased interaction with Sth1, but to a less extent, compared with F220R (Figure 2d and Supplementary Figure 2c). It is expected since Taf14 F220 mediates the hydrophobic contacts with Sth1, and the mutation to a small hydrophobic Ala residue only decreased the hydrophobic contacts but not as disruptive as the arginine mutation.

(2) For the in vivo experiments, the authors work with the Sth1-M4 mutation, and claim that this disrupts specifically the Taf14 interaction. However, this could also affect other proteins (ET domains?) binding to the same motif. Also, could the authors please quantify the WB in Figure 2E ? It looks as if the M4 mutant was already a bit less in the input.

We agree with the reviewer that Sth1-M4 mutation may also affect other ET-containing proteins. In budding yeast, there are four proteins containing ET domains, including Taf14, Sas5, Bdf1, and Bdf2. We already showed in Figure 6d that Bdf1 and Bdf2 did not bind Sth1. We have tried to purify Sas5 ET domain, but unfortunately, Sas5 ET is easy to get aggregated and we cannot get the correctly-folded Sas5 ET domain for biochemical assays (see reference figure 1). Thus at this point, we cannot completely rule out the possibility that Sth1-M4 may also affect binding with other proteins.

We thank the suggestion of WB quantification. We now have included the quantification of WB bands by ImageJ (Figure 2e). Moreover, we included another batch of CoIP

experiment in the Supplementary Figure 2e to show the consistent results that Sth1 M4 mutant decreases the interaction with Taf14_{ET}.

Reference Figure 1

(3) To be better able to judge the quality of the data, it would be essential to show the original ITC curves for the experiments in Figure 4A, not only the final K_d values; only some of them are shown in Figure 4C/D

We are sorry to omit the original plots due to the limited space. Now we have included all the original plots of the reported ITC experiments in Supplementary Figure 4d,e,g,h,i.

(4) For the phase separation (LLPS) assays, the authors should provide more controls. My first concern is the Taf14 protein they use - does it still carry the GST tag, and what GFP variant are they using? 'Canonical' GFP tends to dimerize, and can undergo phase separation by itself, especially in strongly crowding conditions like 10% PEG. Second, I am wondering about the addition of the EBM factors; this could also be additional protein crowding. As a control, they should use RFP only (and whatever else is attached to these peptides).

We thank the reviewer's suggestion and have included more controls in the revision.

We have included the GFP and RFP controls to show that these fluorescence tags do not form phase separation up to 25 μ M concentration in our crowding condition (Figure 5a and Supplementary Figures 5a-b). We have also included controls that adding RFP alone to GFP-Taf14_{ET} did not affect the phase separation (Figures 5f and 5g).

Minor points:

(1) In the introduction, it might be nice to introduce Sth1 with a sentence or two, since it will play a major role in this paper.

Thank you for this suggestion. We have revised the introduction to address why we choose Sth1-Taf14 interaction as an example.

(2) In Supplementary Figure 2A, is it possible that the labelling of the ITC curves (for I1208A and L1210A) was mixed up?

Thanks for good eyes to catch this labeling error. We have corrected it.

(3) For gene expression analysis, the authors use RNA sequencing and see differences when comparing the transcriptome of the wild-type versus the Sth1-M4 carrying strain. The experiments performed seem valid to me. I am aware that this might go beyond the scope of this manuscript, but the ultimate proof of their model would be to demonstrate that Taf14 or RSC recruitment is affected at the genes where they see changes.

This is absolutely correct. We are also interested to see how the Taf14-Sth1 interaction guides the recruitment of RSC to specific gene loci to affect gene expression, which will be studied in a separated manuscript. We have put some discussion on this point in the discussion.

(4) See major comment 4: a list of plasmids / expression constructs would be useful.

In the revision, we have included new Supplementary tables 1 and 2 to list all the strains and plasmids used in this manuscript.

(5) I find the structure in Figure 2C a bit hard to read, especially which amino acids interact. Maybe they can find a clearer perspective, or show two alternative ones ?

We have updated Figure 2c, and hope it will be much easier to read.

(6) In Figure 6E, a scale bar would be helpful.

We are sorry to the missing scale bars. We have included the error bars in Figure 6e, and also checked other figures to ensure all microscopy pictures with scale bars.

(7) At some selected points, the English grammar was not yet perfect, it might be good to send it to a native speaker for a quick read-through.

Thanks. The manuscript has been professionally edited by a native speaker.

Reviewer #4 (Remarks to the Author):

This manuscript reported that the extra-terminal domain of Taf14 recognizes a common motif in multiple transcriptional coactivator proteins from diverse nuclear complexes, and the authors demonstrate that Taf14ET could form liquid-liquid phase separation (LLPS) in vitro. This manuscript not only identifying the molecular mechanism of Taf14ET-related transcriptional mechanisms, but helps us to understand the diverse cellular processes that Taf14ET regulating. However, the functions of Taf14-Sth1 interaction have not fully studied. I believe it is important for this manuscript, this is also why we investigate the transcriptional machineries of Taf14. So, some improvements might need to take into consideration before publication. In this case, the decision of this manuscript is major revision.

Thanks for the constructive suggestions.

Some comments and suggestions:

Major comments:

1. Why the authors use 'Sth1' as an example? I suggest the author explain the reason in Introduction section.

Thank you for this suggestion. We have revised the introduction to address why we choose Sth1-Taf14 interaction as an example.

2. Line 116: In figure 1B, the authors mentioned 'GST-Taf14174-244 could pull-down Sth11183-1359, Sth11183-1240, and 587 Sth11199-1225, but not Sth11248-1359.' But in

Line 116, the author reported that ‘the Sth1 fragments comprising residues 1183-1240 and 1199-1225 were able to associate with Taf14_{ET}’. Why the Sth11183-1359 is lost?

We are sorry not stating this clearly. Now the related text has been changed to “We first confirmed that Sth1₁₁₈₃₋₁₃₅₉ indeed directly interacted with GST-Taf14_{ET} through GST pull-down assay (Figure 1B). Then we generated a series of Sth1 constructs in the residues 1183-1359 range to assess their interactions with GST-Taf14_{ET}. We found that the bromodomain of Sth1 (Sth1₁₂₄₈₋₁₃₅₉) did not interact with Taf14_{ET}, while the Sth1 fragments comprising residues 1183-1240 and 1199-1225 were able to associate with Taf14_{ET} (Figure 1B)”.

3. In Line 120, the authors mentioned that ‘the interaction between Taf14_{ET} and Sth11183-1240 is dominated by hydrophobic interactions rather than electrostatic contacts’, but I do not know why the authors indicated that ‘Electrostatic interactions further stabilize the Sth1EBMC-Taf14_{ET} interaction.’? So, the interaction was dominated by hydrophobic and electrostatic interactions?

We are sorry for the vague statement. We want to emphasize that, although both hydrophobic and electrostatic interactions are required for high-affinity binding between Taf14_{ET} and Sth1₁₁₈₃₋₁₂₄₀, the hydrophobic contacts are more important than electrostatic interactions in mediating Taf14-Sth1 interaction. That’s why we initially used word “dominate”. To avoid confusion, we have changed the related text to the following: “Additionally, the hydrophobic contacts might play an essential role in mediating the Taf14_{ET}-Sth1₁₁₈₃₋₁₂₄₀ complex formation, because increasing the salt concentrations only slightly decreased the observed binding affinity and did not affect enthalpy-change (ΔH)”.

4. Line 139: If we deleted the 9-residue core fragment, and determined the binding capacity? If so, we can further confirm that this motif directly interacts with Taf14_{ET} through mutagenesis study.

Thanks for this suggestion. We did not try to delete the 9-residue core fragment because we were not sure whether such a deletion would affect protein integrity or overall folding. Instead, we used the point mutations and revealed that M4 mutation already

disrupted the Taf14-Sth1 interaction (Figure 2d,2e). We think the point mutation is a better control than the deletion mutation to prove this motif directly interacts with Taf14_{ET}.

5. Why the authors choose temperatures as the phenotype's experiments? I suggest the authors choose different types of stress conditions to confirm the universality of the conclusion. Such as various pH, osmotic pressure. In Figure 3A, the authors did not study the phenotype at 32 oC, how to ensure that the strain grew at 32 oC and below? Figure 3B, how to determine the appropriate concentration of the DNA-damage stress conditions? It is difficult to displayed that no sensitivity to MMS was detected in figure 3B. The author is advised to adjust the contrast of the figure or find a suitable dilution gradient.

We thank the reviewer for raising this concern. First, we are sorry for the typo "32°C". It should be 30°C.

We tested strain sensitivities at different temperatures because previous studies showed *taf14* knockout strains have temperature sensitivity phenotypes. So at the first step, we want to confirm whether the Taf14-Sth1 interaction is important for the cold or heat-resistance response.

According to the reviewer's suggestion, we have performed additional phenotypes experiments in different osmotic pressure. The data are presented in Figure 3b and Supplementary Figure 3b. The *sth1*^{M4} strains failed to grow in the presence of high salt concentration (1 M and 1.5 M NaCl), but generally grew normally at 0.2 M and 0.5 M NaCl condition (Figure 3b). These phenotypes were in sharp contrast to *taf14Δ* and *taf14*^M cells, which exhibited strong growth defects at all tested NaCl concentrations (Supplementary Figure 3b).

Regarding the question "appropriate concentration of the DNA-damage stress conditions", we now present the phenotype data at different concentrations of DNA-damage reagents (Figure 3C and Supplementary Figure 3C). Especially in the presence of MMS, the *taf14*^M mutant strains showed delayed growth at 0.075% and 0.01%, while *sth1*^{M4} strains still showed similar growth phenotypes as WT strains at higher MMS

concentration (0.015% and 0.02%), confirming that *sth1^{M4}* strains is not sensitive to MMS.

6. Line 219: Did the authors select 7 genes for verification? If so, based on what? In addition, the expression levels of these genes in the RNA-seq experiments should also be marked in Figure 3D.

We randomly selected a number of genes (~20 genes) for qPCR validation of RNA-seq data. We chose the genes with a change level more than two-fold shown by RNA-seq. Due to space limitation, we only included seven of them in the main figure in the original submission. In the revision, we have included the remaining qPCR results in the Supplementary Figure 3e. According to the reviewer's suggestion, we have included the change levels from RNA-seq along with qPCR data.

7. Line 227: I do not think the authors have well studied the metabolic pathways that Taf14-Sth1 interaction participated.

We agree with this comment. We just gave an indication that Taf14-Sth1 might be important for metabolic pathways, but did not specify which pathway and how Taf14-Sth1 affected metabolism. This will be an interesting topic for future studies. In the revised manuscript, we have mentioned this point in the discussion section.

8. Line 239: where are the results of Taf2? I have not found the corresponding figure. We are sorry for the typo. Because a previous study already mapped a short peptide (1381-1407) of Taf2 interacted with Taf14 (*JBC* 291,22721-22740, 2016), we did not use GST pull-down to map the Taf14-binding site of Taf2. We directly used ITC to confirm this Taf2 fragment could bind to Taf14_{ET} (Supplementary Figure 4e).

Minor comments:

1. Line 63: 'Saccharomyces cerevisiae' changed to 'S. cerevisiae', please check other expression.

Thanks. We have corrected it.

2. Line 93: as an example

Thanks. We have corrected it.

3. Line 98: 'analysis' to 'analyses'.

Thanks. We have corrected it.

4. Line 119: dissociation constant (Kd); '55 nM'

Thanks. We have corrected it.

5. Line 125: 'nuclear magnetic resonance (NMR)' Abbreviations should be marked when they first appear. In addition, the author needs to check other abbreviations. Some abbreviations need to be marked with their full names.

Thanks. We have checked the manuscript and ensured all abbreviations with full names marked when they first appear.

6. Line 201: 'sth1' Small case

Thanks. We have corrected it.

7. Line 227: there is no figure 3H, is it figure 3F?

Thanks. We have corrected it.

8. Line 261: In Supplementary 4D, is there any control group?

Thanks. Now we have included the control group on Supplementary Figure 4f (original 4d).

9. Line 287: 'Taf14ETs'

Thanks. We have corrected it.

10. Line 336: Please marked the input and output in figure 6D.

Thanks. We have updated Figure 6d to mark the input and output.

REVIEWERS' COMMENTS:

Reviewer #1 (Remarks to the Author):

The authors have made numerous changes to the revised manuscript that clarify their findings and resolve most of the questions raised in the previous round of review. The manuscript is greatly improved and I have no other recommendations or suggested revisions for this submission.

Reviewer #2 (Remarks to the Author):

The authors made efforts to faithfully address all reviewers comments. These efforts reinforced the conclusions made by the authors. As such, it is this reviewers opinion that the manuscript should be accepted for publication.

Reviewer #3 (Remarks to the Author):

I would like to thank the authors for addressing all my points, and congratulations on a beautiful manuscript!

Reviewer #4 (Remarks to the Author):

The author revised the manuscript well, and the current manuscript has been greatly improved. Besides some minor issues, I would in principle recommend publication.

Minor comments:

1. For Supplementary Figure 3d: Error bars were largely missing in the presentation of the experimental data. I could hardly find the error bars from this figure.
2. Figure 3b, Line 472, 492, 549, 550, 570, 610, 629: There should be a space between the unit and the value, please check the manuscript and revise them.
3. Line 259: I suggest the author to unify the form with 'Figure.' or 'Fig.' in the manuscript.
4. Line 495, 485: unify the form with ' μL .' or ' μl '

Ling Jiang

Reviewer #4 (Remarks to the Author):

The author revised the manuscript well, and the current manuscript has been greatly improved. Besides some minor issues, I would in principle recommend publication.

Minor comments:

1. For Supplementary Figure 3d: Error bars were largely missing in the presentation of the experimental data. I could hardly find the error bars from this figure.

Thanks. The Supplementary Fig.3d has been updated in the revised version to include the error bars.

2. Figure 3b, Line 472, 492, 549, 550, 570, 610, 629: There should be a space between the unit and the value, please check the manuscript and revise them.

Thanks. The manuscript has been thoroughly examined to fix these errors.

3. Line 259: I suggest the author to unify the form with 'Figure.' or 'Fig.' in the manuscript.

Thanks. We have unified the manuscript with "Fig." according to NC format guidance.

4. Line 495, 485: unify the form with 'µL.' or 'µl'

Thanks. We have unified the manuscript with 'µL' and 'mL'.